# ACE2-lentiviral transduction enables mouse SARS-CoV-2 infection and mapping of receptor interactions

**Daniel J. Rawle** [1]*, **Thuy T. Le** [1], **Troy Dumenil** [1], **Kexin Yan** [1], **Bing Tang** [1], **Wilson Nguyen** [1], **Daniel Watterson** [2,3], **Naphak Modhiran** [2], **Jody Hobson-Peters** [2,3], **Cameron Bishop** [1], **Andreas Suhrbier** [1,3]

1 Immunology Department, QIMR Berghofer Medical Research Institute, Brisbane, Queensland, Australia,
2 School of Chemistry and Molecular Biosciences, University of Queensland, St Lucia, Queensland, Australia, 3 Australian Infectious Disease Research Centre, GVN Center of Excellence, Brisbane, Queensland, Australia

* Daniel.Rawle@qimrberghofer.edu.au

**Data Availability Statement:** All raw sequencing data (fastq files) are available from the Sequence Read Archive (SRA), BioProject accession:

## Abstract

SARS-CoV-2 uses the human ACE2 (hACE2) receptor for cell attachment and entry, with mouse ACE2 (mACE2) unable to support infection. Herein we describe an ACE2-lentivirus system and illustrate its utility for *in vitro* and *in vivo* SARS-CoV-2 infection models. Transduction of non-permissive cell lines with hACE2 imparted replication competence, and transduction with mACE2 containing N30D, N31K, F83Y and H353K substitutions, to match hACE2, rescued SARS-CoV-2 replication. Intrapulmonary hACE2-lentivirus transduction of C57BL/6J mice permitted significant virus replication in lung epithelium. RNA-Seq and histological analyses illustrated that this model involved an acute inflammatory disease followed by resolution and tissue repair, with a transcriptomic profile similar to that seen in COVID-19 patients. hACE2-lentivirus transduction of IFNAR[-/-] and IL-28RA[-/-] mouse lungs was used to illustrate that loss of type I or III interferon responses have no significant effect on virus replication. However, their importance in driving inflammatory responses was illustrated by RNA-Seq analyses. We also demonstrate the utility of the hACE2-lentivirus transduction system for vaccine evaluation in C57BL/6J mice. The ACE2-lentivirus system thus has broad application in SARS-CoV-2 research, providing a tool for both mutagenesis studies and mouse model development.

## Author summary

SARS-CoV-2 uses the human ACE2 (hACE2) receptor to infect cells, but cannot infect mice because the virus cannot bind mouse ACE2 (mACE2). We use an ACE2-lentivirus system *in vitro* to identify four key amino acids in mACE2 that explain why SARS-CoV-2 cannot infect mice. hACE2-lentivirus was used to express hACE2 in mouse lungs *in vivo*, with the inflammatory responses after SARS-CoV-2 infection similar to those seen in human COVID-19. Genetically modified mice were used to show that type I and III interferon signaling is required for the inflammatory responses. We also show that the hACE2-

PRJNA701678. All other data is available within the paper and supporting information files.

**Funding:** We thank Clive Berghofer and the Brazil Family Foundation (and many others) for their generous philanthropic donations to support SARS-CoV-2 research at QIMR Berghofer MRI (https://www.qimrberghofer.edu.au/) (intramural grants awarded to A.S. and D.J.R). The project was also partly funded by an intramural seed grant from the Australian Infectious Diseases Research Centre (https://aidrc.org.au/) awarded to A.S. A.S. holds an Investigator grant from the National Health and Medical Research Council (NHMRC, https://www.nhmrc.gov.au/) of Australia (APP1173880). The funders had no role in study design, data collection and analysis, decision to publish, or preparation of the manuscript.

**Competing interests:** The authors have declared that no competing interests exist.

lentivirus mouse model can be used to test vaccines. Overall this paper demonstrates that our hACE2-lentivirus system has multiple applications in SARS-CoV-2 and COVID-19 research.

## Introduction

Severe acute respiratory syndrome coronavirus 2 (SARS-CoV-2) has spread rapidly into a global pandemic [1]. SARS-CoV-2 infection can be asymptomatic, but is also the etiological agent of coronavirus disease 2019 (COVID-19), with acute respiratory distress syndrome (ARDS) representing a common severe disease manifestation [2]. The SARS-CoV-2 pandemic has sparked unprecedented global research into understanding mechanisms of virus replication and disease pathogenesis, with the aim of generating new vaccines and treatments. Key to these efforts has been the development of animal models of SARS-CoV-2 infection and COVID-19 disease [3].

The receptor binding domain (RBD) of the spike glycoprotein of SARS-CoV-2 binds human Angiotensin-Converting Enzyme 2 (hACE2) as the primary receptor for cell attachment and entry [4]. Mice do not support productive virus replication because the SARS-CoV-2 spike does not bind to mouse ACE2 (mACE2) [5]. Expression of hACE2 in mice via a transgene allows SARS-CoV-2 infection and provides mouse models that recapitulate aspects of COVID-19. In such models hACE2 is expressed under control of various promoters, including K18 [6–13], mACE2 [14, 15], HFH4 [16], or chicken β-actin [17]. These mouse models all differ in various aspects including level of virus replication, disease manifestations and tissue tropisms [3], but do not provide a simple mechanism whereby genetically modified (GM) or knock-out mice can be exploited for SARS-CoV-2/COVID-19 research. Two systems that do allow the latter, involve transient hACE2 expression in mouse lungs using adenovirus vectors (Ad5) or adeno-associated virus (AAV), which impart the capacity for productive SARS-CoV-2 replication in mouse lungs [10, 18, 19]. A third model involves use of mouse adapted SARS-CoV-2 [20–22], although use of this mutated virus may complicate evaluation of interventions targeting human SARS-CoV-2 strains. However, newly emerged beta (B.1.351) and gamma (P.1) variants, and to a lesser extent the alpha (B.1.1.7) variant, can replicate in C57BL/6J mouse lungs [23].

Lentivirus-mediated gene expression in mouse lung epithelium has been investigated as a treatment for cystic fibrosis and provides long term expression of the cystic fibrosis transmembrane conductance regulator (CFTR) [24–30]. These studies demonstrated that VSV-G pseudotyped lentiviruses can transduce mouse airway and alveolar epithelial cells and their progenitors, resulting in long-term consistent gene expression for up to 24 weeks, suggesting no significant effects of anti-vector immunity on transgene expression. Prolonged transgene expression after lentiviral transduction may represent an advantage over Ad5 transduction systems, where transgene expression in lungs can wane within 2 weeks [31]. Herein we describe an ACE2-lentiviral system that conveys SARS-CoV-2 replication competence *in vitro* and allows productive infection of mouse lungs *in vivo*. We illustrate that GM mice can thereby be accessed for SARS-CoV-2/COVID-19 research, with experiments using IFNAR$^{-/-}$ and IL-28RA$^{-/-}$ mice showing the importance of type I and III IFN responses for SARS-CoV-2-induced inflammation. We also illustrate the use of hACE2-lentiviral transduction of lungs of wild-type C57BL/6J mice for evaluation of vaccines.

## Results

### hACE2-lentiviruses for evaluating the role of RBD binding residues

The hACE2 coding sequence (optimized for mouse codon usage) was cloned into the dual promoter lentiviral vector pCDH-EF1α-MCS-BGH-PGK-GFP-T2A-Puro (herein referred to as pCDH) (Fig 1A), which had been modified to contain the full length EF1α promoter. Previous studies have shown that the EF1α promoter effectively drives gene expression after VSV-G pseudotyped lentivirus transduction of mouse lungs [30]. The GFP marker and puromycin resistance gene (cleaved from GFP by T2A) are expressed from a separate 3-phosphoglycerate kinase (PGK) promoter (Fig 1A). VSV-G pseudotyped pCDH-hACE2 lentivirus was produced by co-transfection of the lentiviral vector with VSV-G and Gag-Pol-Rev plasmids into HEK293T cells. The subsequent lentivirus was used to transduce HEK293T (human embryonic kidney) cells followed by puromycin selection. HEK293T cells do not express hACE2 [32] and therefore do not support SARS-CoV-2 replication, whereas HEK293T cells transduced with hACE2 supported significant virus replication (Fig 1B). A T92Q amino acid substitution in hACE2 was introduced to determine if removing the N90 glycosylation motif increased SARS-CoV-2 replication, as enhanced affinity for spike RBD has been shown by biolayer interferometry assays [33, 34]. hACE2-N90 is glycosylated in HEK293 cells, and when hACE2 was de-glycosylated by endoglycosidase H treatment there was no impact on binding to spike [35], which supports our results that indicate no difference in virus replication for cells expressing hACE2-T92Q (Fig 1B). We also introduced K31N/K353H and T27Y/L79Y/N330Y amino acid substitutions into hACE2 as these changes were predicted by computer modeling to reduce and enhance affinity for spike RBD [33], respectively. HEK293T cells expressing these aforementioned hACE2 amino acid substitutions did not significantly affect SARS-CoV-2 replication (Fig 1B), illustrating that *in silico* affinity predictions do not always translate to significant differences in virus replication *in vitro*.

### Characterizing residues responsible for the inability of mACE2 to support SARS-CoV-2 replication

The ACE2-lentivirus system was used to introduce amino acid substitutions into mACE2 to identify the amino acids that are responsible for restricting SARS-CoV-2 replication in mACE2-expressing cells. Analysis of crystal structures and species susceptibilities have suggested residues that may be responsible for the differences in binding of the RBD to hACE2 and mACE2 [36–40]. Based on these *in silico* studies, we identified seven residues potentially responsible for the human-mouse differences. mACE2-lentivirus vectors were then generated that contained all seven substitutions (N30D/N31K/T79L/S82M/F83Y/A329E/H353K), with two further vectors constructed to identify the minimum requirements for SARS-CoV-2 replication. The latter contained a subset of the seven changes; four amino acid changes (N30D/N31K/F83Y/H353K) and two amino acid changes (N31K/H353K).

As expected, HEK293T cells expressing mACE2 did not support productive SARS-CoV-2 replication, and cytopathic effects (CPE) were not observed. In contrast, all three mACE2 mutants significantly rescued virus replication and infection resulted in CPE (Fig 1C and 1D). However, virus replication was significantly lower for cells expressing mACE2 with only two substitutions (N31K/H353K) (Fig 1C), and infection produced less CPE (Fig 1D). Thus two amino acid changes (N31K and H353K) in mACE2 were sufficient to support SARS-CoV-2 replication; however, at least four changes (N30D/N31K/F83Y/H353K) were required for virus replication and CPE to be comparable to cells expressing wild-type hACE2. These studies highlight the utility of the ACE2-lentivirus system for studying the requirement of ACE2

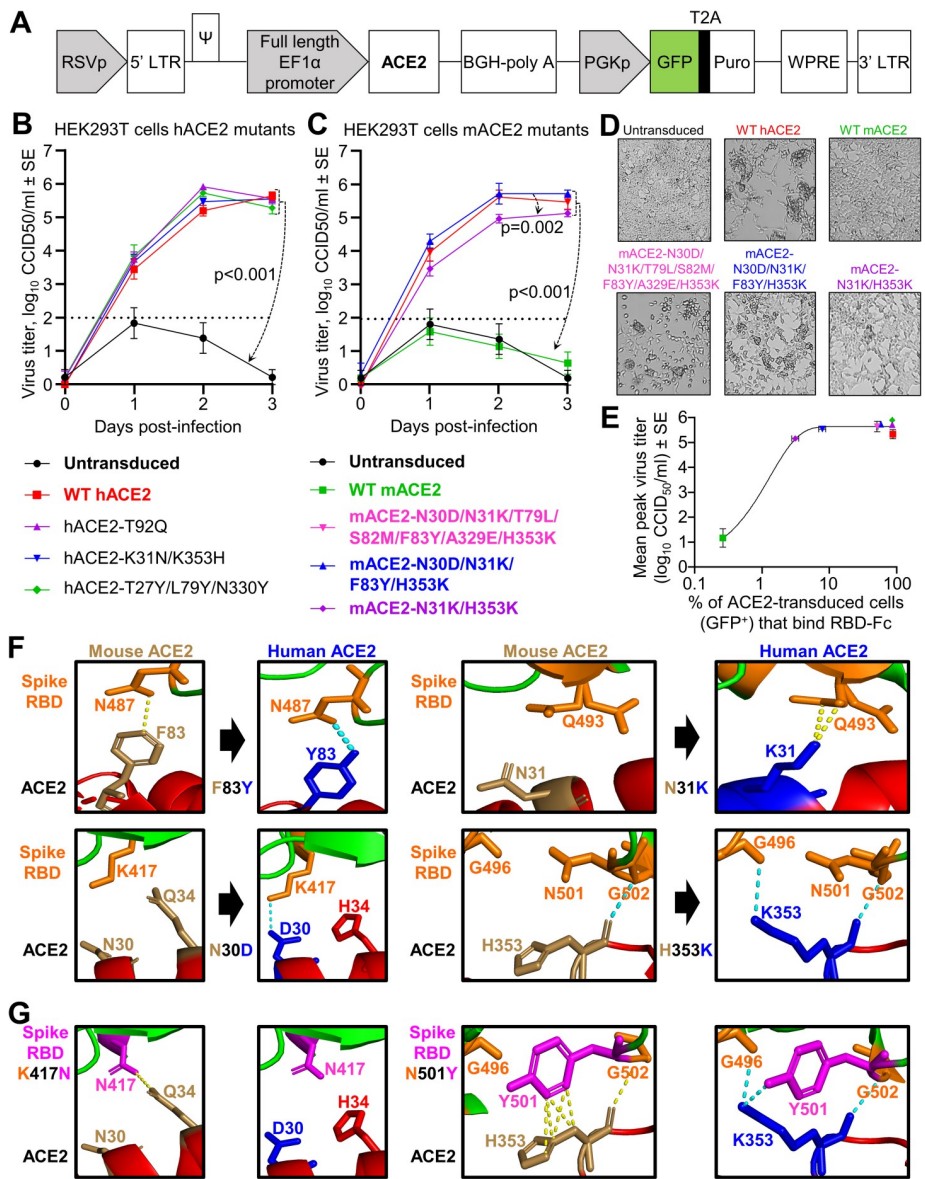

**Fig 1. Mutational analyses of human and mouse ACE2 using ACE2-lentiviruses *in vitro*. A**) Schematic of pCDH-EF1α-ACE2-BGH-PGK-GFP-T2A-Puro lentiviral vector. **B-C**) Growth kinetics of SARS-CoV-2 over a three day time course in HEK293T cells transduced with WT or mutant hACE2 (**B**) or mACE2 (**C**) infected at MOI = 0.1. Data is the mean of 2–4 independent experiments with 2–3 replicates in each and error bars represent SEM. Statistics was determined using Repeated Measures Two-Way ANOVA comparing mock transduced (for B) or WT mACE2 (for C) with all others, and comparing mACE2-N30D/N31K/F83Y/H353K with mACE2-N31K/H353K. **D**) Inverted light microscopy images of HEK293T cells transduced with the indicated ACE2 lentivirus and infected with SARS-CoV-2 at MOI = 0.1. Images were taken at day 3 post infection and were representative of triplicate wells. **E**) SARS-CoV-2 spike RBD-Fc flow cytometry-based binding assay with ACE2-transduced cell lines. The percentage of ACE2-transduced cells (GFP+) with RBD-Fc bound (x-axis) (n = 2 with an additional independent experiment showing similar results) is plotted against mean peak viral titer (y-axis) from 'B' or 'C'. Symbol colors and shapes match the legend presented in 'B' or 'C' for all cell lines. A sigmoidal curve was fitted in GraphPad Prism 8.2.1. See S2 Fig for gating and GFP+ and RBD-Fc+ percentages. **F**) Crystal structure of the spike RBD:hACE2 complex (PBD: 6M0J) [115] viewed in PyMOL and zoomed in on key interactions between ACE2 residues identified in 'C-D' (hACE2 = blue, mACE2 = brown) and spike RBD residues (orange). Yellow dotted lines represent any contacts between chains within 3.5Å, and blue dotted lines represent polar contacts. Direction of black arrows indicate predicted enhanced interactions. **G**) K417N and N501Y mutations (magenta) were introduced in the spike RBD to mimic the beta (B.1.351) and alpha (B.1.1.7) SARS-CoV-2 variants.

residues for productive SARS-CoV-2 infections. A recent study showed that a different mACE2 double mutant (F83Y, H353K) significantly increased RBD binding efficiency and allowed SARS-CoV-2 replication in a mACE2-Ad5 mouse model [41].

## RBD binding assays indicate low levels of receptor binding allow virus replication

To determine whether the ACE2 mutants affected binding to spike RBD, a flow cytometry-based binding assay was performed using the established cell lines and recombinant SARS-CoV-2 spike RBD-Fc protein purified by Protein-A affinity chromatography from RBD-Fc plasmid transfected Chinese Hamster Ovary (CHO) cells. SDS-PAGE followed by Coomassie Brilliant Blue staining illustrated the purity of the recombinant RBD-Fc (S1A Fig), and ELISAs indicated that the RBD-Fc was correctly folded and reactive with an anti-SARS spike specific monoclonal antibody [42, 43]) (S1B Fig). ACE2 transduced cell lines were incubated with RBD-Fc, and binding assessed by anti-human IgG Fcγ-Alexa Fluor 594. Cells were analyzed by flow cytometry, with GFP fluorescence indicating ACE2-lentivirus transduction, and Alexa Fluor 594 fluorescence indicating RBD binding to ACE2 (S2 Fig).

Mock transduced cells showed negligible GFP expression or RBD binding, while all ACE2-transduced cell lines showed GFP expression levels in 67–82% of cells (S2 Fig). The percentage of ACE2-transduced cells (GFP$^+$) that bound RBD-Fc was used as a measure of binding efficiency. WT hACE2 cells showed an 88.34% binding efficiency, while WT mACE2 cells showed a 0.26% binding efficiency (Figs 1E and S2). Cells expressing hACE2-T92Q and hACE2-T27Y/L79Y/N330Y had a RBD binding efficiency similar to WT hACE2, while hACE2-K31N/K353H showed only 7.86% RBD binding (Figs 1E and S2). The binding efficiency for cells expressing mACE2-N31K/H353K was 3.14%, mACE2-N30D/N31K/F83Y/H353K was 58.08%, and mACE2-N30D/N31K/T79L/S82M/F83Y/A329E/H353K was 51.40% (Figs 1E and S2). When plotted against peak virus titer from growth kinetics assays (Fig 1B and 1C), it was clear that low levels of RBD:ACE2 binding in this flow cytometry-based binding assay were sufficient for efficient virus replication (Fig 1E).

## Modeling the RBD:ACE2 key interactions for mouse, human and virus variants

Of all the potential interactions between the RBD and ACE2 previously identify by X ray crystallography, and of all the amino acid differences between human and mouse ACE2 [36, 40], our data identified a key role for 4 ACE2 amino acid residues for conferring SARS-CoV-2 infectivity on mACE2-expressing cells. Modeling of the interactions between these 4 residues and the RBD predict improved interactions between the virus and the receptor (Fig 1F), providing an explanation for the gain of function for the mutated mACE2. The F83Y substitution replaced a non-polar interaction between mACE2-F83 and RBD-N487, with a polar interaction between hACE2-Y83 and RBD-N487. The N31K substitution created new interactions between ACE2-K31 and RBD-Q493. The N30D substitution created a salt bridge interaction between ACE2-D30 and RBD-K417. Lastly, the H353K substitution created a new hydrogen bond with RBD-G496.

Recently, SARS-CoV-2 variants have emerged with mutations in the RBD, with two of the key mutations involving interactions with hACE2 residues 30 and 353 [44]. Beta variants (B.1.351) have the RBD mutations K417N and N501Y, and the alpha variants (B.1.1.7) have the N501Y mutation [44]. Modelling predicts that the K417N mutation would result in the gain of an interaction between RBD N417 and mACE2-Q34, whereas the interaction between K417 and hACE2-D30 would be lost (Fig 1G). The selection for RBD-K417N in humans may

therefore be more related to antibody escape [45, 46], rather than improving interactions with hACE2. The N501Y mutation in the RBD is predicted to generate new interactions between Y501 with mACE2-H353, and new polar interactions between Y501 and hACE2-K353 (Fig 1G). Our modeling supports the observation that viruses with K417N and N501Y mutations have increased affinity for mACE2, with both appearing in mouse-adapted SARS-CoV-2 [47]. This is supported by a recent study which shows the RSA variant has robust replication in mouse lung, while the alpha (B.1.1.7) variant has weaker replication [23].

## hACE2-lentivirus transduction of mouse cell lines imparts SARS-CoV-2 replication competence

To determine whether hACE2-lentivirus transduction could make mouse cell lines SARS-CoV-2 replication competent and thus available for SARS-CoV-2 research, 3T3 (mouse embryonic fibroblasts) and AE17 (mouse lung mesothelioma cell line) cells were transduced with hACE2. Significant virus replication was seen in transduced cells (S3 Fig), although somewhat lower than that seen in transduced HEK293T cells. Overt CPE was not seen (S3 Fig). Host cell proteases including furin, transmembrane protease serine 2 (TMPRSS2) and cathepsin B and L (CatB/L), are involved in cleaving SARS-CoV-2 spike and this is important in cell entry [48–50]. Differences in expression of these host proteases may contribute to different replication efficiency between cell lines, although differences in other host proteins may also be involved. Overall, this illustrates that the hACE2-lentivirus system can be used for mouse cell lines, but that the efficiency of viral replication may be cell line or cell type dependent.

## hACE2-lentivirus transduction of C57BL/6J, IFNAR[-/-] and IL-28RA[-/-] mouse lungs

To illustrate the utility of the hACE2-lentivirus system for use in GM mice, we investigated the role of type I and type III IFN receptor signaling in SARS-CoV-2 infections. C57BL/6J, IFNAR[-/-] and IL-28RA[-/-] were inoculated into the lung via the intranasal (i.n.) route with 1.2–2.2×10^4 transduction units of hACE2-lentivirus, equivalent to approximately 190–350 ng of p24 per mice (Fig 2A), or vehicle (DMEM with 5% FBS). An hour prior to administration of lentivirus or vehicle, all mice received intrapulmonary lysophosphatidylcholine (LPC) (1%) to enhance transduction efficiency [27, 51]. One week later, mock transduced and hACE2 transduced mice received an intrapulmonary challenge with SARS-CoV-2 ($10^5$ $CCID_{50}$ per mouse), and lungs were collected at 2, 4 and 6 days post infection for C57BL/6J and IFNAR[-/-] mice and day 2 for IL-28RA[-/-] mice (Fig 2A). A one week interval between hACE2 transduction and challenge was chosen because previous studies indicate this is peak transgene expression, with no change in expression between 1 and 24 weeks post lentivirus transduction of mouse lungs [24, 25]. Mice did not display overt clinical symptoms or weight loss (Fig 2B). RT-qPCR analyses indicated similar levels of hACE2 mRNA after transduction of mouse lungs in each of the 3 mouse strains, with expression maintained over the 6 day course of the experiment at levels significantly higher than mock transduced mice (Fig 2C). hACE2 or SARS-CoV-2 mRNA was not detected in mouse brains (S4 Fig). RNA-Seq read counts for hACE2 (S5A Fig) closely matched RT-qPCR results (Fig 2C), indicating good constancy between these two assays.

## SARS-CoV-2 replication was similar between hACE2-lentivirus transduced C57BL/6J, IFNAR[-/-] and IL-28RA[-/-] mouse lungs

Lungs from infected C57BL/6J, IFNAR[-/-] and IL-28RA[-/-] mice were analyzed for infectious virus tissue titers. Mock transduced lungs showed no detectable virus titers (Fig 2D), whereas

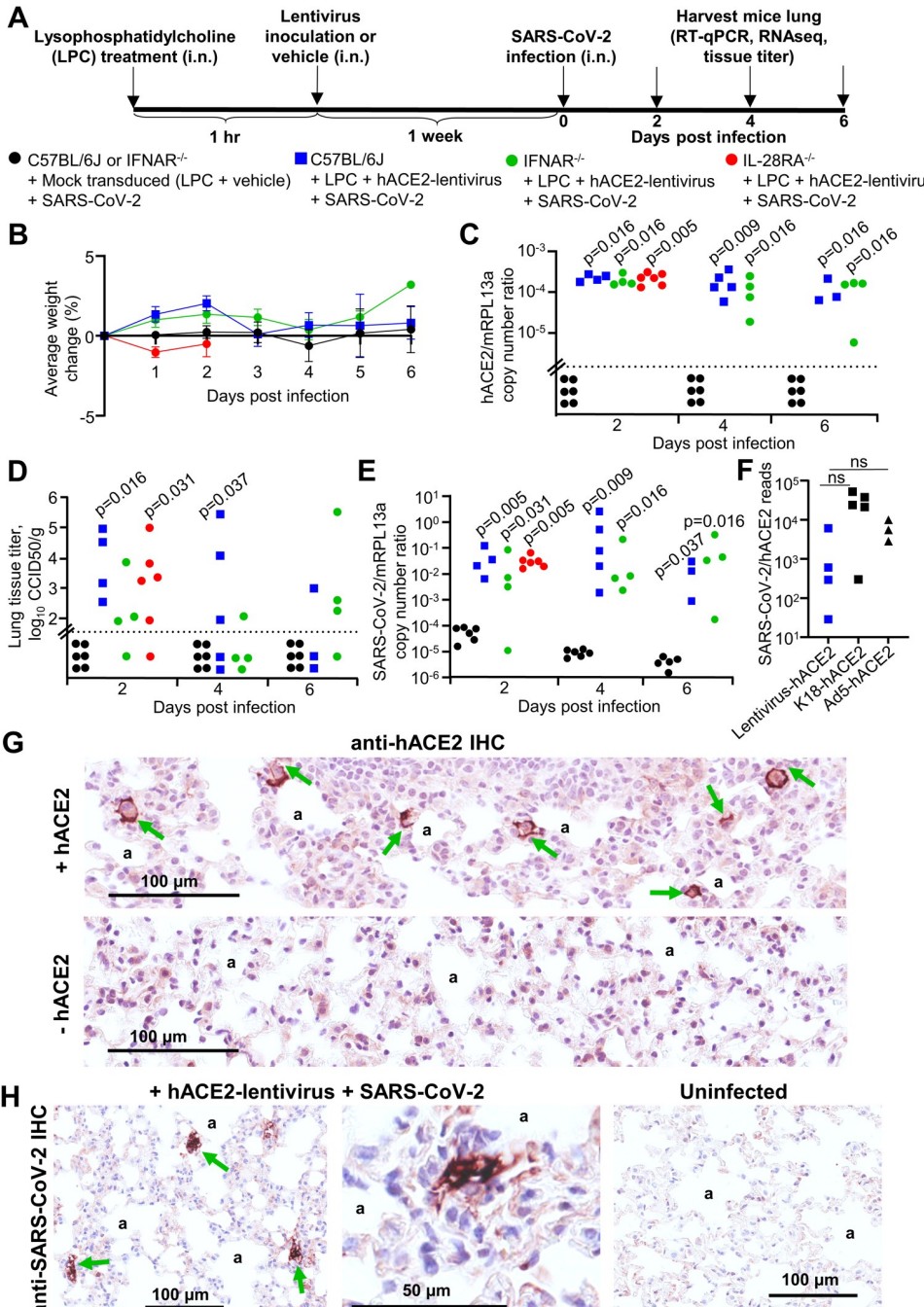

**Fig 2. SARS-CoV-2 replication in hACE2-lentivirus transduced C57BL/6J, IFNAR[-/-] and IL-28RA[-/-] mouse lungs.**
**A)** Timeline of lentivirus transduction of mouse lungs and SARS-CoV-2 infection. **B)** Percent weight change for mice in C-E (mock transduced mice, n = 6 at each time point. C57BL/6J mice, n = 4 at day 2, n = 5 at day 4, n = 3 at day 6. IFNAR[-/-] mice, n = 4 at each time point. IL-28RA[-/-], n = 6 at day 2). Data represents the mean percent weight loss from day 0 and error bars represent SEM. **C)** RT-qPCR of mouse lungs RNA using primers for hACE2 (introduced by lentivirus transduction) normalized to mRPL13a levels. Data is for individual mice and is expressed as RNA copy number calculated against a standard curve for each gene. Horizontal line indicates cut-off for reliable detection, with all mock transduced mice falling below this line. Statistics are by Kolmogorov Smirnov test compared to mock transduced samples. **D)** Titer of SARS-CoV-2 in mouse lungs determined using $CCID_{50}$ assay of lung homogenates. Horizontal line indicates the limit of detection of 1.57 $\log_{10}$ $CCID_{50}$/g. Statistics are by Kolmogorov Smirnov test compared to mock transduced mice. IFNAR[-/-] versus mock transduced mice at day 2 post infection reaches significance by Kruskal-Wallis test (p = 0.018). **E)** RT-qPCR of mouse lung RNA using primers for SARS-CoV-2 E

gene normalized to mRPL13a levels. Data is for individual mice and is expressed as RNA copy number calculated using a standard curve. Statistics are by Kolmogorov Smirnov test compared to mock transduced mice. C57BL/6J hACE2-transduced versus mock transduced mice at day 4 post infection reaches significance by Kruskal-Wallis test (p = 0.046). See S6 Fig for SARS-CoV-2 RNA copies normalised to hACE2 copies. **F)** SARS-CoV-2 read counts (also see S5 Fig) normalized to hACE2 read counts from RNA-seq data for lentivirus-hACE2 transduced mice at day 2, K18-hACE2 transgenic mice at day 4, and downloaded data for Ad5-hACE2 transduced mice at day 2 [53]. Not significant by Kolmogorov Smirnov or Kruskal-Wallis tests. **G)** Examples of anti-hACE2 IHC (brown staining, green arrows) in mouse lung alveolar epithelial cells adjacent to alveolar spaces ('a'). Images are representative of lung sections from at least 6 mice and additional images are shown in S7 Fig. **H)** Examples of anti-SARS-CoV-2 IHC (brown staining, green arrows) in mouse lung alveolar epithelial cells adjacent to alveolar spaces ('a'). Images are representative of lung sections from at least 6 mice.

hACE2-lentivirus transduced lungs showed significant viral titers ranging from $10^2$ to $10^5$ $CCID_{50}/g$ for all strains of mice on day 2, with titers dropping thereafter (Fig 2D). Viral RNA levels were measured using RT-qPCR, with mock transduced lungs showing low and progressively decaying levels post-inoculation (Fig 2E). For all hACE2-lentivirus transduced lungs, significantly elevated and persistent viral RNA levels were seen for all strains and at all time-points by RT-qPCR (Fig 2E). Reducing viral titers (Fig 2D) despite lingering viral RNA levels (Fig 2E) have been reported previously for K18-hACE2 mice [9] and has also been suggested in human infections [52]. RNA-Seq read counts for SARS-CoV-2 (S5B Fig) closely matched RT-qPCR results (Fig 2E). As might be expected, viral RNA levels (measured by RT-qPCR or titer) showed a highly significant correlation with hACE2 RNA expression levels (S6A and S6B Fig).

Analysis of the RNA-Seq data showed that there were fewer total SARS-CoV-2 and hACE2 reads in infected hACE2-lentivirus transduced mice when compared (i) with infected K18-hACE2 transgenic mice, and (ii) with publically available data for infected hACE2-Ad5 transduced mice [53] (S5D–S5G Fig). However, when SARS-CoV-2 reads were normalised to hACE2 reads, there were no significant differences [53] (Fig 2F), indicating receptor expression levels largely dictate the levels of SARS-CoV-2 replication in these different systems.

Importantly, no significant differences in viral loads emerged between C57BL/6J, IFNAR$^{-/-}$ and IL-28RA$^{-/-}$ mice (Fig 2D and 2E). This remained true even when viral RNA levels were normalized to hACE2 mRNA levels (S6C Fig). Similar results were obtained by RNA-Seq, with viral read counts not significantly different for the 3 mouse strains (S5A–S5C Fig). Thus using three different techniques, no significant effects on viral replication could be seen in mice with type I or type III IFN receptor deficiencies.

## hACE2 and SARS-CoV-2 localize to alveolar epithelial cells in hACE2-lentivirus transduced mouse lungs

To analyze localization of hACE2 expression in mouse lungs after hACE2-lentivirus transduction, immunohistochemistry (IHC) using an anti-hACE2 specific antibody was performed. The IHC illustrated that after hACE2-lentivirus transduction, hACE2 expression was primarily localized to alveolar epithelial cells (Figs 2G and S7). This observation is consistent with the literature that suggests VSV-G pseudotyped lentiviruses transduce alveolar epithelial cells more readily than airway epithelial cells [54].

According to the supplier, the antibody used herein (Abcam, ab108209) does not react with mouse or rat ACE2. This antibody stained bronchiolar epithelial cells, alveolar septa, alveolar macrophages and the endothelium of blood vessels after Ad5-hACE2 transduction of mouse lungs [10]. In contrast, in situ hybridization (FISH) in this model suggested hACE2 RNA expression was restricted to alveolar epithelium [18]. Humanized mice with hACE2 inserted by CRISPR and driven by the mACE2 promoter showed limited hACE2 staining, which was

largely localized to airway epithelium [15]. Other anti-hACE2 antibodies used in related studies [11, 14] cross-react with mouse ACE2 according to the suppliers.

An anti-SARS-CoV-2 spike monoclonal antibody was used in IHC to determine localization of virus antigen in infected mouse lungs at day 2. The IHC illustrated that virus replication was localized to alveolar epithelium (Fig 2H), consistent with hACE2 staining (Fig 2G). Overall, IHC indicated that hACE2 and SARS-CoV-2 localized to alveolar epithelial cells, which is similar to localization reported in an Ad5-hACE2 mouse model [18]. Importantly, alveolar epithelial cells are the major target of SARS-CoV-2 infection and are implicated in COVID-19 in humans [55, 56].

## SARS-CoV-2 replication induces histopathological lesions

Leukocyte infiltration and bronchiolar membrane sloughing have previously been described as histopathological features of SARS-CoV-2 infection in K18-hACE2 [9, 57] and Ad5-hACE2 mouse lungs [18, 19, 53]. These features were examined in H&E stained lung sections from hACE2-lentivirus transduced mice and compared with mock transduced mice, with both groups receiving an intrapulmonary inoculum of SARS-CoV-2. There were more and larger areas of leukocyte infiltration in +hACE2 mice compared to -hACE2 mice, which reached significance when day 2 and 6 data were combined (Fig 3A). Leukocyte infiltrates were predominantly located in perivascular regions, with some in peribronchiolar regions (Fig 3B). Bronchiolar membrane sloughing was also significantly increased at day 6 in +hACE2 mice compared to -hACE2 mice (Fig 3B and 3C).

Bronchiole occlusion and oedema, smooth muscle hyperplasia and alveolar thickening (measured by white space) [9, 57] were also analyzed. Although the former approached significance, the remaining features did not show significant differences (S8A–S8D Fig), with viral inoculation even in the absence of active replication able to induce lung pathology.

## SARS-CoV-2 replication in hACE2-lentivirus transduced mouse lungs induces inflammatory signatures by day 2 post-infection

To determine the innate responses to SARS-CoV-2 replication in hACE2-transduced mouse lungs, gene expression in lungs of infected hACE2-transduced mice was compared with infected mock transduced mice on day 2 post-infection using RNA-Seq. Differentially expressed genes (DEGs) between these groups were identified using a false discovery rate (FDR, or q value) threshold of <0.05 (Fig 4A and S1A–S1C Dataset). Of the 110 DEGs, 95 were upregulated, with ≈40% of these classified as type I IFN-stimulated genes (ISGs) as classified by Interferome (using conservative settings that only included genes validated *in vivo* for mouse) (Fig 4B, full list in S1D Dataset). Type III ISGs are known to be poorly annotated in this database, and so this analysis likely under-estimates the number of such ISGs. Three of the most upregulated DEGs (Ccl8, Ccr3 and Cd163) were validated by RT-qPCR, and were not significantly induced by LPC + lentivirus without SARS-CoV-2 infection (S9A Fig). There were also no obvious inflammatory lesions (such as areas of leukocyte infiltrate) in H&E stained lungs from mice that received intrapulmonary hACE2-lentivirus, when compared with naïve mice (S9B Fig). This argues that the RNA-Seq-derived transcription signatures are predominantly due to SARS-CoV-2 replication rather than arising from lentivirus transduction.

Ingenuity Pathway Analysis (IPA) of the 110 DEGs produced a series of pro-inflammatory Upstream Regulators (USRs) (S1E and S1F Dataset) and included Th1, Th2 and Th17-associated cytokine USRs, as well as type I IFN USRs (Fig 4C). SARS-CoV-2-specific T cells in humans were reported to be predominantly Th1-associated, although Th2-associated cytokines were also identified [58]. DEG lists for cytokine and type I IFN responses to SARS-CoV-

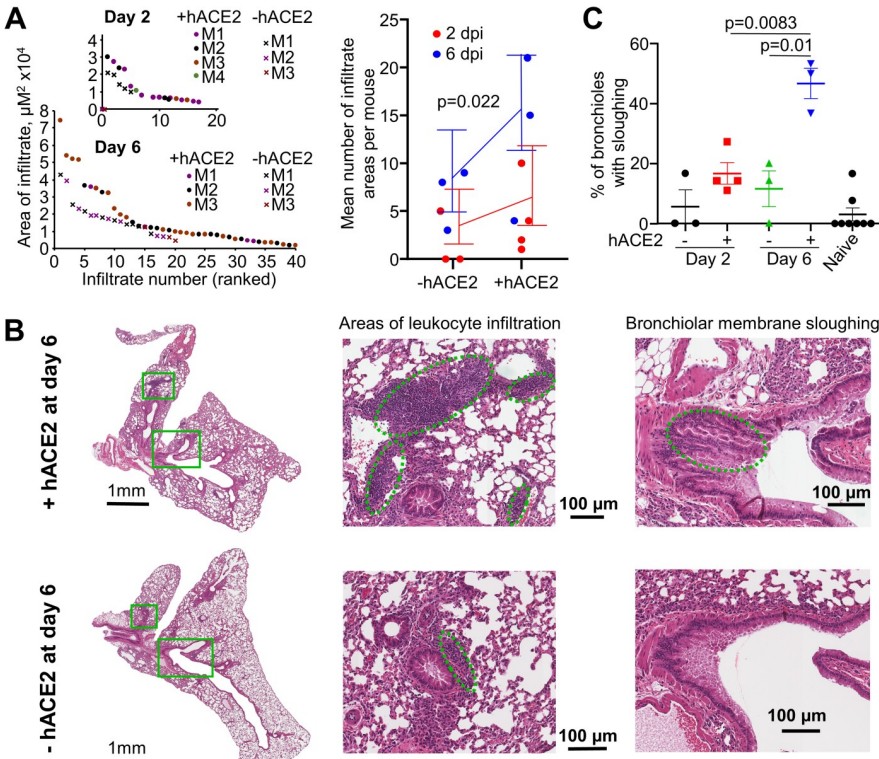

**Fig 3. SARS-CoV-2 replication in hACE2-transduced mouse lungs increased leukocyte infiltration and bronchiolar membrane sloughing. A)** The areas of dense infiltrates (lesions) in lung sections from 3–4 mice per group, at day 2 and 6 were measured and plotted, ranked largest to smallest lesion (left). More and larger lesions were seen in mice whose lungs had been transduced with hACE2 (+hACE2) and infected with an intra-lung inoculum of SARS-CoV-2, than in mock transduced lungs (-hACE2) that had also received an intra-lung inoculum of SARS-CoV-2. Statistics (right) used both day 2 and 6 and modelled lesion counts with a Poisson generalised linear model, using the square root of the sum of lesions sizes for each mouse to weight each mean. Error bars represent 95% confidence intervals. **B)** Representative images of H&E stained lung sections on day 6 for mice that received hACE2-lentivirus transduction and infected with an intra-lung inoculum of SARS-CoV-2 (+hACE2), and mice that received transduction that had also received an intra-lung inoculum of SARS-CoV-2 (-hACE2). Examples of areas of leukocyte infiltration and bronchiolar membrane sloughing are indicated by dashed green ovals. **C)** Bronchiolar membrane sloughing was counted when there was evidence of bronchiole epithelial cells shedding from the membrane into the lumen. The percentage of bronchioles with evidence of sloughing as a proportion of total bronchioles is shown. n = 3–4 mice per group, n = 8 for naïve mice. Statistics was determined using t-tests.

2 in infected K18-hACE2 transgenic mouse lungs compared to uninfected lungs have been published [9]. These DEG lists (S1G Dataset) were used to interrogate the full pre-ranked (by fold change) gene list (Fig 4A and S1B Dataset) by Gene Set Enrichment Analysis (GSEA). Significant enrichments were observed (Fig 4D), indicating similar cytokine and type I interferon (IFN) responses after SARS-CoV-2 infection of K18-hACE2 transgenic and hACE2-lentivirus transduced mice.

IPA *Diseases and Functions* analysis of the 110 DEGs revealed a series of annotations dominated by cellular infiltration and innate responses signatures (Fig 4E and S1H and S1I Dataset). Several of the annotations were associated with monocytes and macrophages (Fig 4E), consistent with a previous report showing that inflammatory monocyte-macrophages were the major source of inflammatory cytokines in mouse adapted SARS-CoV infected mice [59]. The cellular infiltrates in COVID-19 patient lungs are also dominated by monocyte-macrophages with a suggested role in severe disease [60]. Inflammatory signaling was also the dominant

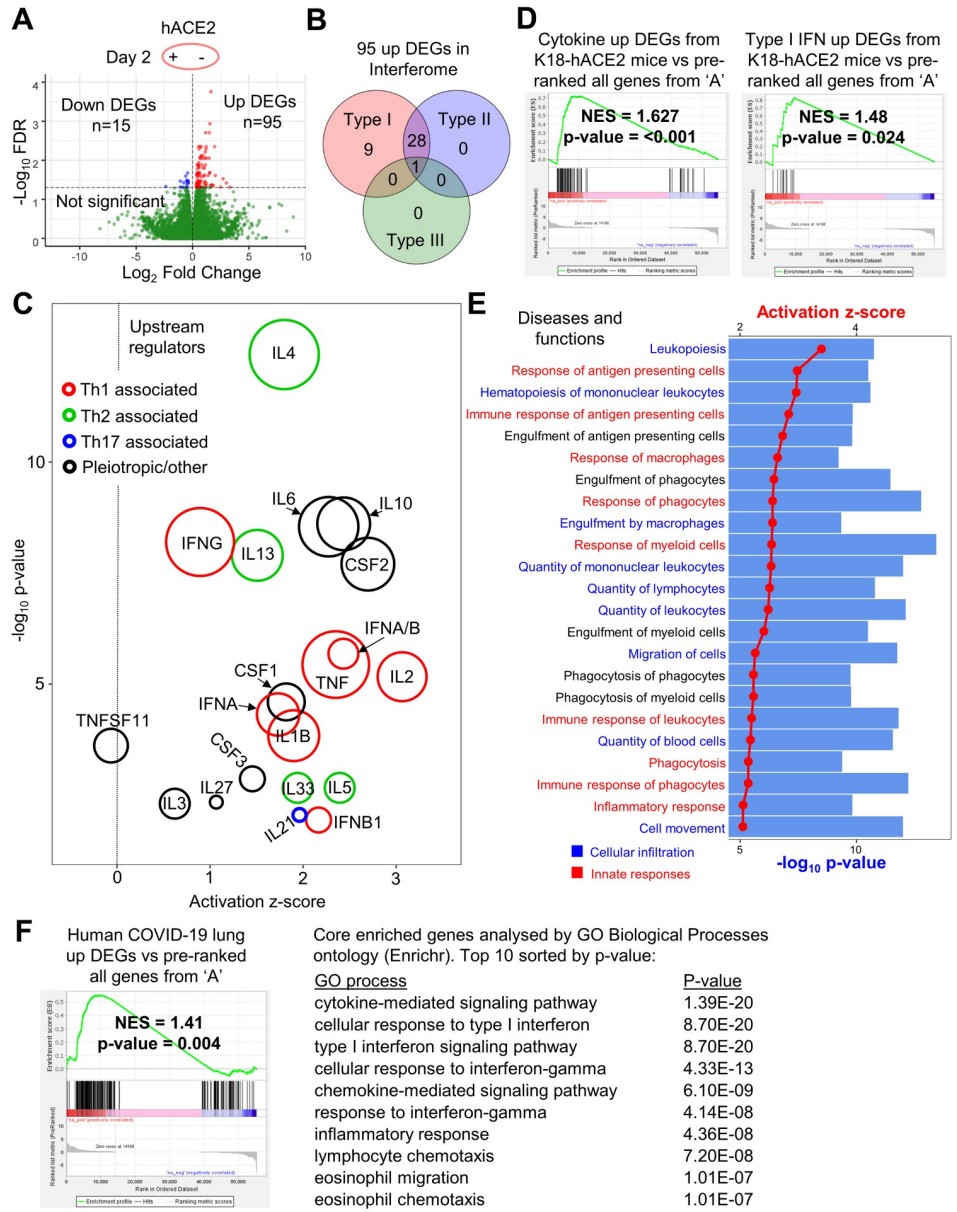

**Fig 4. SARS-CoV-2 replication in hACE2-lentivirus transduced mouse lung induces inflammatory signatures by day 2 post-infection. A)** Volcano plot of gene expression from RNA-seq analyses of lungs at day 2 comparing mice with and without hACE2-lentivirus transduction. Genes were considered DEGs if they had a FDR value < 0.05 (above horizontal dotted line) (see S1B–S1C Dataset for full gene and DEG lists). **B)** Interferome analysis of 95 up DEGs from 'A'. 38 of 95 up DEGs (40%) were ISGs. Red = type I IFN, blue = type II IFN, green = type III IFN (see S1D Dataset for full Interferome ISG list). **C)** Cytokine signatures identified by IPA USR analysis (see S1E and S1F Dataset for full and cytokine only USR lists) of 110 DEGs identified in 'A'. Circle area reflects the number of DEGs associated with each USR annotation. USRs were grouped into three categories; red = Th1 associated, green = Th2 associated, blue = Th17 associated, and black = pleiotropic/other. The vertical dotted line indicates activation z-score of 0. **D)** GSEAs for cytokine-related DEGs or type I IFN-related DEGs (S1G Dataset) from Winkler et al. supplemental data [9] against the pre-ranked all gene list (S1B Dataset). Winkler et al. compared infected versus uninfected K18-hACE2 transgenic mouse lung at day 2 post infection by RNA-Seq and provided DEG lists for these categories in their supplementary material. Normalised enrichment score (NES) and nominal p-value are shown. **E)** IPA diseases and functions analysis (see S1H–S1I Dataset for full lists) of 110 DEGs identified in 'A'. The 23 annotations with the most significant p-value and with an activation z-score of >2 were plotted with dots/lines indicating activation z-score and bars indicating–log₁₀ p-value. Annotations were grouped into two categories; cellular infiltration (blue) and innate responses (red). **F)** GSEA for DEGs from human COVID-19 lungs versus healthy control from Blanco-Melo et al. [64] against the pre-ranked (by fold change) all gene list (S1B Dataset). Normalised enrichment score (NES) and nominal p-value are

shown. Core enriched genes (see S1M Dataset for full core enriched gene list) determined by GSEA were entered into Enrichr and the top 10 GO Biological Processes annotations sorted by p-value are shown (see S1N Dataset for full GO processes list).

signature in other analyses including GO Biological Processes and Cytoscape (S1J and S1K Dataset).

GSEA analysis using the >31,000 gene sets available from the molecular signatures database (MSigDB), indicated significant negative enrichment (negative NES scores) of gene sets associated with translation and mitochondrial electron transport chain function (S1L Dataset). Translation inhibition by SARS-CoV-2 Nsp1 via the blocking of mRNA access to ribosomes has been reported previously [61, 62]. SARS-CoV-2 infection-induced down-regulation of genes associated with cellular respiration and mitochondrial function has also been shown in human lungs [63].

DEGs from human COVID-19 lungs [64] were also enriched in our hACE2-lentivirus transduced mouse lungs (Fig 4F), and analysis of the core enriched genes indicated the similarity was due to cytokine, IFN, chemokine and inflammatory signaling genes (Fig 4F and S1M and S1N Dataset). Another human COVID-19 lung dataset was also interrogated [65] and the results again illustrated enrichment of immune-related DEGs in the hACE2-lentivirus transduced mice (S1O Dataset).

Overall these analyses illustrate that SARS-CoV-2 infection in hACE2-lentivirus transduced C57BL/6J mouse lungs significantly recapitulate responses seen in other murine models and in COVID-19 patients.

## Day 6 post-infection is characterized by inflammation resolution and tissue repair signatures

To determine how response profiles progress in the hACE2-lentivirus model, RNA-Seq was used to compare lungs on day 2 with lungs on day 6 post-infection, with day 6 broadly representing the time of severe lung disease in the K18-hACE2 mouse model [9]. As noted above (Fig 2D), the virus titers had dropped in hACE2-lentivirus transduced mouse lungs over this time period by ≈2.8 logs, and severe disease (typically measured by weight loss) was not evident in this hACE2-lentivirus model (Fig 2B), although there were increased leukocyte infiltrates and bronchiolar membrane sloughing at day 6 (Fig 3). In both hACE2-lentivirus transduced and mock transduced C57BL/6J mice there was a clear evolution of responses from day 2 to day 6 post-infection; however, DEGs obtained from the former were largely distinct from DEGs obtained from the latter (Fig 5A). The RNA-Seq data thus illustrated that the DEGs associated with virus infection were distinct from those associated with virus inoculation (Fig 5A). RNA-Seq of infected lungs provided 551 DEGs, 401 up-regulated and 150 down-regulated (Fig 5B and S2A–S2C Dataset).

IPA USR analyses of the 551 DEGs showed a clear reduction in inflammatory cytokine annotations, although the IL4 signature remained a dominant feature on day 6 (Fig 5C and S2D and S2E Dataset). A Th2 skew and IL-4 have both been associated with lung damage in COVID-19 patients [66]. On day 6 there was an up-regulation of USRs primarily associated with tissue repair (Fig 5D and S2F Dataset); MAPK1 (also known as ERK) and MAPK9 [67], BMP7 [68], TCF7L2 [69], and KLF4 [70, 71]. B and T cell-associated USRs were also seen (Fig 5D, green circles) [72–74], consistent with the development of an adaptive immune response. A relatively minor PARP1 signature was identified (Fig 5D), with PARP inhibitors being considered for COVID-19 therapy [75, 76]. A strong signature associated with estrogen receptor 1 (ESR1) was observed (Fig 5D), with estrogen having anti-inflammatory properties and

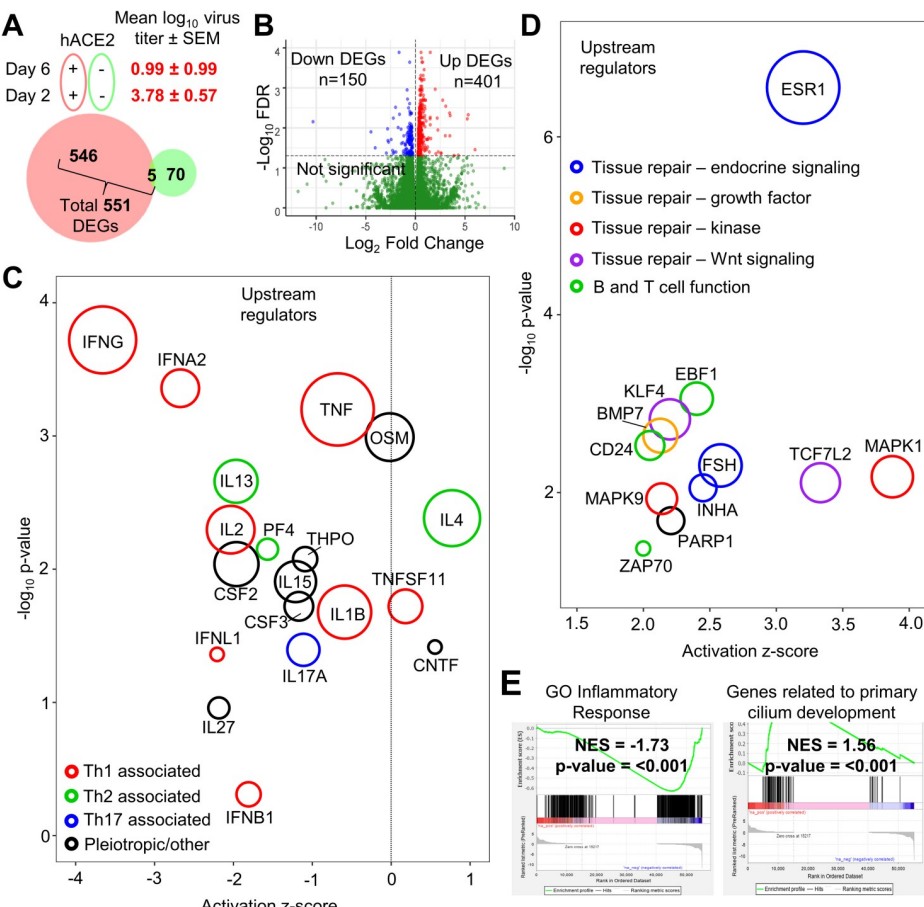

**Fig 5. Day 6 post-infection is characterized by inflammation resolution and tissue repair signatures. A)** Venn diagram for DEGs comparing day 6 versus 2 post infection with (red) and without (green) hACE2-lentivirus transduction. Number of DEGs that are FDR<0.05 are shown. Mean virus titer ($\log_{10}$ CCID$_{50}$/g) in lung tissue ± SEM (represents data shown in Fig 2D) is shown for hACE2 transduced mice at day 2 or 6 post infection. **B)** Volcano plot of gene expression from RNA-seq analyses of hACE2-lentivirus transduced lung RNA comparing day 6 with day 2 (see S2B and S2C Dataset for gene and DEG lists). Genes were considered DEGs if they had a FDR value < 0.05 (above horizontal dotted line). **C)** Cytokine signatures identified by IPA USR analysis (see S2D and S2E Dataset for full and cytokine only lists) of 551 DEGs identified in 'B' (S2C Dataset). Circle area reflects the number of DEGs associated with each USR annotation. USRs were grouped into four categories; red = Th1 associated, green = Th2 associated, blue = Th17 associated, and black = pleiotropic/other. The vertical dotted line indicates activation z-score of 0. **D)** Upregulated USR signatures identified using IPA analysis (see S2F Dataset) of 551 DEGs identified in 'B'. Circle area reflects the number of DEGs associated with each USR annotation. USRs (excluding chemicals) with an activation z-score of >2 are shown and were grouped into five categories; green = B and T cell function, blue = tissue repair–endocrine signaling, orange = tissue repair–growth factor, red = tissue repair–kinase, purple = tissue repair–Wnt signaling. **E)** Selected plots from GSEA of entire MSigDB database using pre-ranked (by fold change) 'all' gene list (S2B Dataset). NES score and p-value is shown for the "GO_inflammatory_response" and "Wp_Genes_Related_To_Primary_Cilium_Development_Based_On_Crispr" annotations.

believed to be associated with reduced COVID-19 severity in women [77–79], and reduced SARS-CoV disease severity in female mice [59, 80]. Follicle stimulating hormone (FSH) (Fig 5D) stimulates estrogen production, and inhibin subunit alpha (INHA) is involved in the negative feedback of FSH production [81]. IPA Diseases and Functions analysis, Cytoscape, and GO Biological Processes analyses further supported the contention that on day 6 inflammation had abated and tissue repair was underway (S2G–S2L Dataset). Histopathological analyses indicated increased cellular infiltrates on day 6 (Fig 3), and this was supported by Diseases and

Functions analyses with annotations related to cellular movement and quantity significantly upregulated (S2G Dataset). Our RNA-Seq analyses would thus suggest that these resolution phase infiltrates are involved in tissue repair.

The >31,000 gene sets available in MSigDB were used to interrogate the complete gene list (pre-ranked by fold change) for day 6 versus day 2 (S2B Dataset) by GSEA. A highly significant negative enrichment for inflammatory response, and a highly significant positive enrichment for cilium development was seen (Fig 5E). Cilia in mammals are found in the lining of the respiratory epithelium. These GSEAs thus provide further support for the conclusion above, together arguing that in the hACE2-lentivirus model, day 6 post infection is characterized by inflammation resolution and tissue repair.

## Type I and III IFN signaling is required for SARS-CoV-2-induced lung inflammation

To determine what responses to SARS-CoV-2 infection are dependent on type I IFN signaling, the lentivirus system was used to analyze IFNAR$^{-/-}$ mouse lungs using RNA-Seq. IFNAR$^{-/-}$ and C57BL/6J mice were transduced with hACE2-lentivirus and were then infected with SARS-CoV-2 and lungs harvested for RNA-Seq on days 2 and 6 post-infection. The largest impact of type I IFN signaling deficiency was seen on day 2 post infection (Fig 6A), consistent with inflammation resolution on day 6 (Fig 5C–5E). The total number of DEGs for IFNAR$^{-/-}$ versus C57BL/6J mouse lungs on day 2 was 192, with most of these down-regulated (Fig 6A and S3A–S3C Dataset). Viral reads were not significantly different between these two mouse strains (Fig 6A), confirming that knocking out type I IFN signaling is insufficient to significantly affect SARS-CoV-2 replication [82].

We showed in Fig 4 that RNA-Seq analysis revealed 110 DEGs associated with inflammatory responses when hACE2-transduced versus mock transduced C57BL/6J mouse lungs on day 2 post infection were compared. When the same comparison was made for hACE2-lentivirus transduced versus mock transduced lungs in SARS-CoV-2 infected IFNAR$^{-/-}$ mice, only 1 DEG (Kcnk5) was identified (Fig 6B). This clearly illustrated that the inflammatory responses on day 2 in C57BL/6J mice (Fig 4C) required intact type I IFN signaling.

As a control, the number of DEGs for IFNAR$^{-/-}$ mice versus C57BL/6J in mock transduced lungs was determined. Although virus replication (hACE2-lentivirus transduced lung) provided 192 DEGs, virus inoculation (mock transduced lung) provided only 16 DEGs (Fig 6C). Thus again, virus inoculation had a limited effect on these analyses, with DEGs largely associated with virus infection and replication.

Type III IFN is thought to act as a first-line of defense against infection at epithelial barriers, with the more potent and inflammatory type I IFN response kept in reserved in the event the first line of defense is breached [83, 84]. As type III IFN signaling has been implicated in reducing SARS-CoV-2 infection *in vitro* [85], the aforementioned analyses on day 2 was repeated in IL-28RA$^{-/-}$ mice. Viral reads were not significantly different between these two mouse strains (Fig 6D), confirming that knocking out type III IFN signaling was insufficient to significantly affect SARS-CoV-2 replication [82, 86]. The DEG list for IL-28RA$^{-/-}$ versus C57BL/6J mice comprised 132 genes (S4A–S4C Dataset), of which 84 were down-regulated (Fig 6D). Perhaps surprisingly, the DEGs were mostly different from that seen in IFNAR$^{-/-}$ mice (Fig 6D), arguing that the ISGs stimulated by type I and III IFNs are largely distinct. The majority (90%) of the 178 down-regulated DEGs in IFNAR$^{-/-}$ versus C57BL/6J mice on day 2 post infection were ISGs as defined by Interferome (using *in vivo* and mouse only settings) (S3D Dataset). Of the 84 down-regulated genes in IL-28RA$^{-/-}$ lungs, only 27% were identified as ISGs (S4D Dataset). However, there are no annotations in the Interferome database for type III ISGs in mice, and

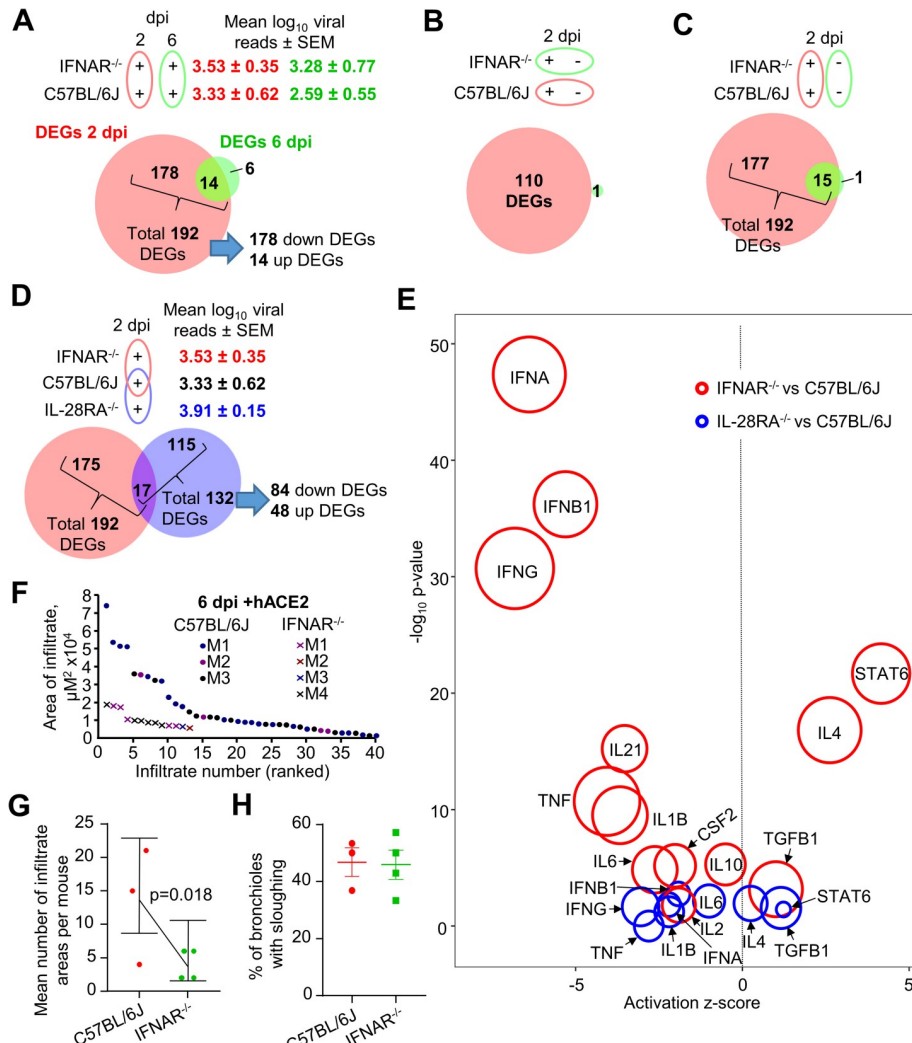

**Fig 6. Type I and III IFN signaling is required for SARS-CoV-2-induced lung inflammation. A)** Venn diagram for number of DEGs (FDR<0.05) between IFNAR[-/-] versus C57BL/6J mice at day 2 (red) or day 6 (green). Mean $\log_{10}$ SARS-CoV-2 read count in RNA-seq data ± SEM is shown (red text = day 2, green text = day 6). **B)** Venn diagram for number of DEGs (FDR<0.05) between plus versus minus hACE2-lentivirus transduction comparing C57BL/6J (red) and IFNAR[-/-] (green) mice at day 2. **C)** Venn diagram for IFNAR[-/-] versus C57BL/6J mice at day 2 comparing hACE2-lentivirus transduced (red) or mock transduced (green) mice. **D)** Venn diagram comparing IFNAR[-/-] versus C57BL/6J (red) and IL-28RA[-/-] versus C57BL/6J (blue) (see S3B and S3C Dataset and S4B and S4C Dataset for full gene and DEG lists) mice at day 2 (all mice had received hACE2-lentivirus transduction). Mean $\log_{10}$ SARS-CoV-2 read count in RNA-seq data ± SEM is shown (red text = IFNAR[-/-], black text = C57BL/6J, blue text = IL-28RA[-/-]). **E)** Signatures identified using IPA USR analysis of DEGs identified in 'D' for IFNAR[-/-] versus C57BL/6J (red) and IL-28RA[-/-] versus C57BL/6J (blue). Circle area reflects the number of DEGs associated with each USR annotation. Signatures were selected focusing on data previously identified in 'Fig 4C' (entire list is in S3E Dataset and S4E Dataset). **F)** The areas of dense infiltrates (lesions) in lung sections from 3–4 mice per group at day 6 were measured and plotted, ranked largest to smallest lesion. Data for 'C57BL/6J' mice is from 'Fig 3A'. More and larger lesions were seen in WT mice compared to IFNAR[-/-] mice. **G)** Statistics modelled lesion counts with a Poisson generalised linear model, using the square root of the sum of lesions sizes for each mouse to weight each mean. Error bars represent 95% confidence intervals. **H)** Bronchiolar membrane sloughing was determined when there was evidence of bronchiole epithelial cells shedding from the membrane into the lumen. The percentage of bronchioles with evidence of sloughing as a proportion of total bronchioles is shown. n = 3–4 mice per group.

changing the settings to include human data (which is also under-annotated for type III IFN) did not increase the number of type III ISGs in our search using down DEGs from IL-28RA[-/-] lungs.

To determine the pathways regulated by type I and III IFN in response to SARS-CoV-2 infection, IPA USR analyses was performed on DEG lists from day 2 for hACE2-lentivirus transduced lungs for IFNAR[-/-] versus C57BL/6J (192 DEGs) and IL-28RA[-/-] versus C57BL/6J (132 DEGs). IPA USR signatures associated with pro-inflammatory cytokine signaling were significantly reduced in both IFNAR[-/-] and, to a lesser extent, IL-28RA[-/-] mice when compared to C57BL/6J mice (Fig 6E, full lists in S3E and S4E Datasets). The USR annotations for type I and III IFN receptor deficiencies were largely similar (Fig 6E), although they arose from a largely distinct set of DEGs (Fig 6D). This likely reflects expression of type I and type III IFN by different cell types with differing response profiles to the same cytokines [83].

Taken together these results illustrate that while there was no significant effects on viral replication, lungs from type I and type III IFN receptor knockout mice both had a significantly blunted acute pro-inflammatory response following SARS-CoV-2 infection.

## Type I IFN signaling drives leukocyte infiltration during SARS-CoV-2-induced lung inflammation

Overt leukocyte infiltrates were observed in this model on day 6 post infection (Fig 3A). H&E staining of lungs showed that hACE2-lentivirus transduced IFNAR[-/-] mice had significantly fewer and smaller leukocyte infiltrate lesions on day 6 post infection when compared with hACE2-lentivirus transduced C57BL/6J mice (Fig 6F and 6G). Levels of bronchiolar membrane sloughing was not significantly different (Fig 6H). IPA Diseases & Functions annotations also indicated lower cellular infiltrates in hACE2-lentivirus transduced IFNAR-/- mice on day 2 post infection (S3F Dataset). Our data is consistent with other studies using AAV and Ad5 hACE2 transduction systems, which similarly showed reduced leukocyte infiltration in IFNAR[-/-] mice [19, 53].

## Immunization with infectious or UV-inactivated SARS-CoV-2 protects against virus infection

To evaluate the utility of the hACE2-lentivirus mouse model of SARS-CoV-2 replication for vaccine studies, C57BL/6J mice were immunized twice subcutaneously (s.c.) with either infectious SARS-CoV-2 or UV-inactivated SARS-CoV-2 [87] formulated with adjuvant (Fig 7A). Significant serum neutralization titers were observed for all mice post-boost (Fig 7B), comparable with the higher end levels seen in people with past SARS-CoV-2 infections [88].

Mice were challenged with intrapulmonary SARS-CoV-2 and sacrificed on day 2. All mouse lungs had detectable hACE2 mRNA by RT-qPCR, with levels higher in unvaccinated mice (Fig 7C), likely due to immune cell infiltrates (see below) diluting hACE2 mRNA. Mice immunized with either infectious or UV-inactivated SARS-CoV-2 showed significant reductions in viral RNA levels by RT-qPCR (Fig 7D) in tissue titrations (Fig 7E) and by RNA-Seq read counts (Fig 7F). (Day 2 data for the mock transduced and unvaccinated C57BL/6J control mice are the same as those presented in Fig 2). These immunizations thus protected mice against significant viral infections and demonstrated the utility of hACE2-lentivirus transduced mice as a model for vaccine evaluation.

Lungs harvested on day 2 were also analyzed by RNA-Seq (S5A and S5B Dataset), with 1376 DEGs identified for infectious virus immunization versus unimmunized mice (DEGs in S5C Dataset). The DEGs were analyzed by IPA canonical pathways, USR analysis, and the Diseases and Functions feature, which all showed infiltration signatures, dominated by T cell

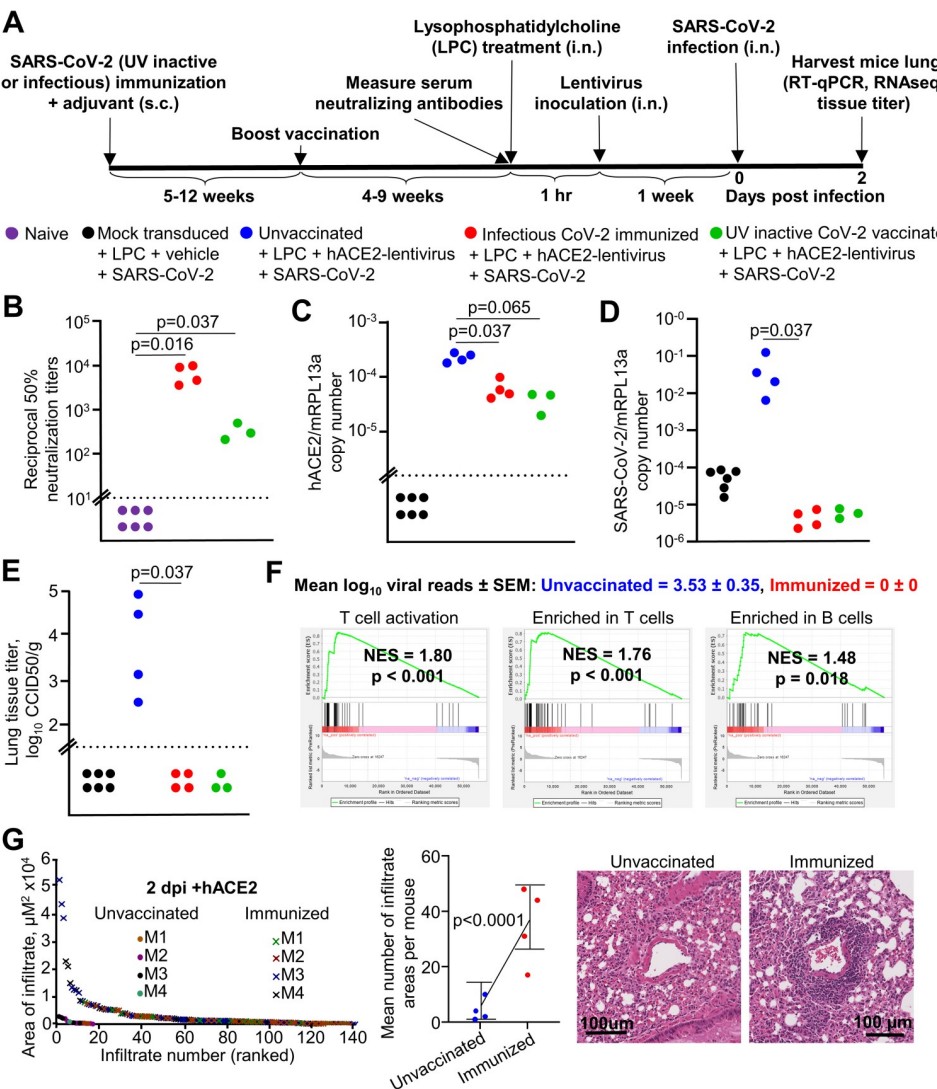

**Fig 7. Immunization with infectious or UV-inactivated SARS-CoV-2 protects against virus infection. A)** Timeline of mice immunization, lentivirus transduction and SARS-CoV-2 infection. Mock transduced mice, n = 6 (data from Fig 2). Unvaccinated C57BL/6J mice, n = 4 (data from Fig 2). Infectious CoV-2 immunized mice, n = 4. UV inactive CoV-2 vaccinated, n = 3 mice. **B)** Reciprocal 50% neutralization titers of naïve mice and infectious or UV SARS-CoV-2 immunized mice. Horizontal dotted line represents the limit of detection (1 in 10). Statistics are by Kolmogorov Smirnov test. **C)** RT-qPCR of mouse lung RNA using primers for hACE2 introduced by lentivirus transduction normalized to mRPL13a levels. Data is individual mice and is expressed as RNA copy number calculated against a standard curve for each gene. Horizontal line indicates cut-off for reliable detection, with all hACE2-negative mice falling below this line. Statistics are by Kolmogorov Smirnov test. **D)** RT-qPCR of mouse lungs RNA using primers for SARS-CoV-2 E gene normalized to mRPL13a levels. Data is for individual mice and is expressed as RNA copy number calculated against a standard curve for each gene. Statistics are by Kolmogorov Smirnov test. Unvaccinated versus UV-inactive CoV-2 vaccinated reaches significance by Kruskal-Wallis test (p = 0.034). **E)** Titer of SARS-CoV-2 in mouse lungs determined using $CCID_{50}$ assay of lung homogenate. Horizontal line indicates the limit of detection of 1.57 $\log_{10}$ $CCID_{50}$/g. Statistics are by Kolmogorov Smirnov test. Unvaccinated versus UV-inactive CoV-2 vaccinated reaches significance by Kruskal-Wallis test (p = 0.028). **F)** Mean $\log_{10}$ SARS-CoV-2 read count at day 2 post-infection in RNA-seq data ± SEM is shown (blue text = unvaccinated mice, red text = infectious SARS-CoV-2 immunized mice). GSEA for Blood Transcription Modules (BTMs) from Li et al. [89] against the pre-ranked all gene list comparing unvaccinated versus infectious SARS-CoV-2 immunized (s.c) mouse lung at day 2 (all mice had hACE2-lentivirus transduction). Selected BTM modules with a positive NES score and p < 0.05 are shown (full list in S5 Dataset). See S5 Dataset for full gene and DEG lists and full downstream analyses lists. **G)** The areas of dense infiltrates (lesions) in lung sections from unvaccinated and immunized mice (n = 4 each group) at day 2 were measured and plotted, ranked largest to smallest lesion (left). Data for 'unvaccinated' mice is from 'Fig 3A'. More and larger lesions were seen in immunized mice compared to unvaccinated mice. Statistics modelled lesion counts with a Poisson generalised linear

model, using the square root of the sum of lesions sizes for each mouse to weight each mean (middle). Error bars represent 95% confidence intervals. Examples of perivascular infiltrates in H&E staining of lung sections from unvaccinated and immunized mice at day 2 post-infection (right).

annotations (S5D–S5F Dataset). This was confirmed by GSEA using human blood transcription modules [89] against the pre-ranked (by fold change) gene list for immunized versus unvaccinated mice (S5G Dataset). This showed the expected enrichment of T cell and B cell annotations in immunized mice (Fig 7F), and illustrated that protection against virus replication in lungs of immunized mice was associated with a significant infiltration of lymphocytes. H&E staining also showed significantly more infiltrates in protected mice, primarily at perivascular sites (Fig 7G). Our results are consistent with studies in COVID-19 patients, which show that early SARS-CoV-2-specific T cell responses promote rapid viral clearance leading to milder disease [90].

## Discussion

Herein we describe a hACE2-lentivirus transduction system that allows the use of C57BL/6J and GM mice for SARS-CoV-2/COVID-19 research. Virus titers in hACE2-lentivirus transduced mouse lung were comparable to other studies using mice transduced with Ad5 or AAV expressing hACE2 [10, 18, 19]. These transduction methodologies avoid the fulminant brain infection seen in K18-hACE2 mice, with infection of this organ the major contributor to mortality in the K18-hACE2 model [91, 92]. Brains of mice that received intrapulmonary hACE2-lentivirus inoculation had no detectable hACE2 or SARS-CoV-2 RNA, consistent with literature that suggests there is no transduction of olfactory epithelium following intrapulmonary lentivirus inoculation in mice [93], with microinjection directly into the olfactory epithelium required for transduction [94]. The key advantage of lentiviral over adenoviral systems is the integration of the lentiviral payload into the host cell genome, thereby providing stable expression. This would allow, for instance, the use of GM mice for studies of re-infection after recovery from initial infection. Ad5-hACE2 mice had significantly reduced hACE2 expression 4 days after infection [18], and thus would likely need re-administration of Ad5-hACE2 for re-infection studies. The hACE2-lentivirus thus provides a new option for studying SARS-CoV-2 infection that can be readily applied using standard lentivirus technology. We characterize the lentivirus-hACE2 model of SARS-CoV-2 infection in C57BL/6J mice, and illustrate the use of lentivirus-hACE2 to assess the role of type I and type III IFNs in virus control and inflammatory responses using IFNAR$^{-/-}$ and IL-28RA$^{-/-}$ mice, respectively. We also illustrate the use of C57BL/6J mice transduced with lentivirus-hACE2 for SARS-CoV-2 vaccine evaluation.

The lentivirus-hACE2 C57BL/6J mouse model of SARS-CoV-2 infection shares many cytokine signatures with those seen in human COVID-19 patients; IL-2, IL-10, IL-6, TNFα, IL-4, IL-1β, IFNγ and CSF3 signatures were found in SARS-CoV-2-infected lentivirus-hACE2 transduced C57BL/6J mice on day 2 post infection, and elevated levels of these cytokines are found in human COVID-19 patients [95, 96]. Several of these cytokines have been implicated as mediators of COVID-19; for instance, IL-6 [97–99], IL-4 [66], IL-10 [100] and IFNγ [101]. On day 6 post infection the lentivirus-hACE2 C57BL/6J mouse model showed a series of signatures associated with tissue repair. Thus our RNA-Seq analyses illustrated that this model involves an acute inflammatory disease followed by resolution and tissue repair. The hACE2-lentivirus model is thus a model of mild disease, comparable with some Ad5-hACE2 models [10].

hACE2-lentivirus transduction in IFNAR$^{-/-}$ and IL-28RA$^{-/-}$ mice was used to confirm that type I IFN signaling [19] and show that type III IFN signaling are involved in driving

inflammatory responses. Virus replication was not significantly affected in IFNAR[-/-] and IL-28RA[-/-] mice, largely consistent with other studies using different model systems, which indicates that knocking out both pathways (STAT1[-/-], STAT2[-/-], or combined IFNAR[-/-]/IL-28RA[-/-]) is required before an impact on virus replication is seen [19, 82, 86, 102]. A high level of cross-compensation is thus implicated, as is described in influenza infection [84]. *In vitro* experiments show that SARS-CoV-2 replication is sensitive to type I and III IFNs, although infection does not stimulate particularly high levels of IFN [85, 103, 104]. A number of SARS-CoV-2 encoded proteins inhibit IFN production and signaling [98, 105, 106]. In infected cells, such inhibition of signaling would limit the antiviral activity of IFNs, as well as limiting the positive feedback amplification that is needed for high levels of IFN production [107]. IFN therapies have been proposed for COVID-19 patients [108] and such treatment may induce an antiviral state in uninfected cells, thereby perhaps limiting viral spread if given prophylactically or early during the course of infection [109]. However, caution might be warranted for such therapy during later stage COVID-19 as type I and III IFN signaling are shown herein to drive inflammatory responses during SARS-CoV-2 infection, and are also known to have pro-inflammatory activities in other settings [107].

The main limitation of the hACE2-lentivirus model compared to transgenic or adenovirus hACE2 models is lower hACE2 expression levels, resulting in lower and more variable SARS-CoV-2 infection ($\approx$2.5 log range in hACE2-lentivirus transduced mice, compared to $\approx$1–2 log range in the K18-hACE2 mice [12, 18, 91], although such variation was also higher in some Ad5-hACE2 models [10]). Lower hACE2 expression levels might be expected given the robust and consistent transgene expression in K18-hACE2 mice, and the lung tropism of adenoviral type-5 vectors. Adenovirus vectors can also be grown to very high titers allowing delivery of up to $2.5 \times 10^8$ plaque forming units per mouse [10, 18, 19, 53], whereas lentivirus production capacity generally limits the dose that can be delivered ($2 \times 10^4$ transduction units was used herein). Another limitation, shared with other transduction based-models, is that LPC + lentivirus transduction can itself induce immune responses, potentially complicating analyses of SARS-CoV-2-infection induced responses. However, we show LPC + lentivirus responses are relatively minor, and are likely also to be lower than adenovirus-vector induced inflammation [110]. Herein we have used LPC + vehicle + SARS-CoV-2 as the negative control, with the absence of hACE2 preventing SARS-CoV-2 replication. However, intrapulmonary inoculation of virus, although unable to replicate, nevertheless results in lung pathology, albeit less severe than that seen in hACE2 transduced mice. Although such an inoculation control ensured that responses identified herein were associated with virus replication, such controls are rarely used in related studies. For instance, analyses in K18-hACE2 mice generally do not use inoculation of UV-inactivated virus as a replication deficient control. Distinguishing responses associated with inoculation from those associated with replication, thus remains an issue for all mouse models where direct delivery of virus into the lungs is required for efficient infection. In humans, infection starting in the upper respiratory tract is thought to travel down to the lower respiratory tract over time [111, 112], a feature not recapitulated in mouse models.

Herein we also illustrate the use of the hACE2-lentivirus system to study the role of ACE2 residues on SARS-CoV-2 replication *in vitro*. Only four "mouse-to-human" changes in mACE2 (N30D, F83Y, N31K, H353K) enabled full SARS-CoV-2 replication and CPE to levels similar to those seen for infection of hACE2-expressing cells. These results suggest a CRISPR mouse with these substitutions would be able to support SARS-CoV-2 infection and provide a model of COVID-19. Remarkably, a minimum of only two substitutions in mACE2 (N31K, H353K), which only marginally increased spike RBD binding, were sufficient for partial restoration of SARS-CoV-2 infection and CPE. Changing the same residues in hACE2 to their mouse equivalents (K31N, K353H) did not affect virus replication compared to WT hACE2,

but significantly impaired interaction with spike RBD. Taken together (Fig 1E), these data illustrate that robust virus replication is achieved *in vitro* despite low binding efficiency in the RBD-Fc flow cytometry-based assay. Each SARS-CoV-2 virion contains approximately $24 \pm 9$ spike trimers non-uniformly distributed on the surface of the particle [96]. Multiple spike-hACE2 interactions may promote high avidity interactions [113, 114], thereby allowing virus entry even when the affinity of RBD-hACE2 interactions is low. Such considerations have implications for deriving predictions of virus replication capacity from binding data or *in silico* affinity calculations.

In conclusion, we describe the development of a mouse model using intrapulmonary hACE2-lentivirus transduction to sensitize mouse lungs to SARS-CoV-2 infection, which recapitulated cytokine signatures seen in other mouse models and COVID-19. We demonstrate broad applicability of this new mouse model in vaccine evaluation, GM mice infection, and *in vitro* evaluation of ACE2 mutants.

## Materials and methods

### Ethics statement and regulatory compliance

All mouse work was conducted in accordance with the "Australian code for the care and use of animals for scientific purposes" as defined by the National Health and Medical Research Council of Australia. Mouse work was approved by the QIMR Berghofer Medical Research Institute animal ethics committee (P3600, A2003-607). For intrapulmonary inoculations, mice were anesthetized using isoflurane. Mice were euthanized using $CO_2$ or cervical dislocation.

Breeding and use of GM mice was approved under a Notifiable Low Risk Dealing (NLRD) Identifier: NLRD_Suhrbier_Oct2020: NLRD 1.1(a). Cloning and use of lentiviral vectors for transduction of ACE2 in to mice lungs and cell lines was approved under an NLRD (OGTR identifier: NLRD_Suhrbier_Feb2021: NLRD 2.1(l), NLRD 2.1(m)).

All infectious SARS-CoV-2 work was conducted in a dedicated suite in a biosafety level-3 (PC3) facility at the QIMR Berghofer MRI (Australian Department of Agriculture, Water and the Environment certification Q2326 and Office of the Gene Technology Regulator certification 3445).

### Cell lines and SARS-CoV-2 culture

Vero E6 (C1008, ECACC, Wiltshire, England; obtained via Sigma Aldrich, St. Louis, MO, USA), HEK293T (a gift from Michel Rist, QIMR Berghofer MRI), Lenti-X 293T (Takara Bio), AE17 (a gift from Dr Delia Nelson, Faculty of Health Sciences, Curtin Medical School), and NIH-3T3 (American Type Culture Collection, ATCC, CRL-1658) cells were cultured in medium comprising DMEM for Lenti-X 293T cells or RPMI1640 for all others (Gibco) supplemented with 10% fetal calf serum (FCS), penicillin (100 IU/ml)/streptomycin (100 μg/ml) (Gibco/Life Technologies) and L-glutamine (2 mM) (Life Technologies). Cells were cultured at 37˚C and 5% $CO_2$. Cells were routinely checked for mycoplasma (MycoAlert Mycoplasma Detection Kit MycoAlert, Lonza) and FCS was assayed for endotoxin contamination before purchase [116]. The SARS-CoV-2 isolate was kindly provided by Dr Alyssa Pyke (Queensland Health Forensic & Scientific Services, Queensland Department of Health, Brisbane, Australia). The virus (hCoV-19/Australia/QLD02/2020) was isolated from a patient and sequence deposited at GISAID (https://www.gisaid.org/; after registration and login, sequence can be downloaded from https://www.epicov.org/epi3/frontend#1707af). Virus stock was generated by infection of Vero E6 cells at multiplicity of infection (MOI)≈0.01, with supernatant collected after 2–3 days, cell debris removed by centrifugation at 3000 x g for 15 min at 4˚C, and virus aliquoted and stored at -80˚C. Virus titers were determined using standard $CCID_{50}$ assays (see

below). The virus was determined to be mycoplasma free using co-culture with a non-permissive cell line (i.e. HeLa) and Hoechst staining as described [117].

## $CCID_{50}$ assays

Vero E6 cells were plated into 96 well flat bottom plates at $2x10^4$ cells per well in 100 μl of medium. For tissue titer, tissue was homogenized in tubes each containing 4 ceramic beads twice at 6000 x g for 15 seconds, followed by centrifugation twice at 21000 x g for 5 min before 5 fold serial dilutions in 100 μl RPMI1640 supplemented with 2% FCS. For cell culture supernatant, 10 fold serial dilutions were performed in 100 μl RPMI1640 supplemented with 2% FCS. 100 μl of serially diluted samples were added to Vero E6 cells and the plates cultured for 5 days at 37˚C and 5% $CO_2$. The virus titer was determined by the method of Spearman and Karber (a convenient Excel $CCID_{50}$ calculator is available at https://www.klinikum.uni-heidelberg.de/zentrum-fuer-infektiologie/molecular-virology/welcome/downloads).

## Lentivirus cloning

ACE2 genes were cloned into pCDH-EF1α-MCS-BGH-PGK-GFP-T2A-Puro Cloning and Expression Lentivector (System Biosciences, catalogue number CD550A-1), where the EF1α promoter was replaced with the full length EF1α using NheI and ClaI restriction enzymes (New England Biolabs). Human ACE2 coding sequence (codon optimized for mouse) was cloned into the pCDH lentivector with PCR fragments amplified using Q5 High-Fidelity 2X Master Mix (New England Biolabs) and the following primers; vector backbone (pCDH amplified with Forward 5'- TAAATCGGATCCGCGG -3' and Reverse 5'- AATTCGAATTCGCTA GC) and hACE2 insert (Forward 5'-GCTAGCGAATTCGAATTATGAGCAGCAGCTCTTG GC -3' and Reverse 5'-CCGCGGATCGCATTTATCAGAAGCTTGTCTGCACGT -3'). The pCDH fragment was digested with DpnI (New England Biolabs) and was purified using QIA-quick Gel Extraction Kit (QIAGEN), as was the hACE2 fragment. Fragments were recombined using NEBuilder HiFi DNA Assembly Cloning Kit as per manufacturers' instructions. This was transformed into NEB 10-beta Competent E. coli (High Efficiency) (New England Biolabs) as per manufacturers' instructions and plated on ampicillin agar plates overnight. Colony PCR was performed using Q5 High-Fidelity 2X Master Mix (New England Biolabs) and the hACE2 insert primers to identify a positive colony, which was grown in LB broth with ampicillin and plasmid was purified using NucleoBond Xtra Midi kit (Machery Nagel).

All ACE2 mutant coding sequences were ordered from Twist Bioscience/Decode Science containing EcoRI upstream and BamHI downstream. Fragments were amplified using the using Q5 High-Fidelity 2X Master Mix (New England Biolabs) and the following primers; Forward 5'- CCTGACCTTAGCGAATTCATG -3' and Reverse 5'- ACCTAGCCTCGCGGATC -3'. PCR fragments were purified using Monarch DNA Gel Extraction Kit (New England Biolabs), and were digested with EcoRI and BamHI (New England Biolabs), as was the pCDH vector before purification again with Monarch DNA Gel Extraction Kit (New England Biolabs). The ACE2 fragment was ligated with pCDH vector using T4 DNA Ligase (New England Biolabs) as per manufacturers' instructions and then transformed into NEB 10-beta Competent E. coli (High Efficiency) (New England Biolabs) as per manufacturers' instructions and plated on ampicillin agar plates overnight. Colonies were grown in LB broth with ampicillin and plasmid was purified using NucleoBond Xtra Midi kit (Machery Nagel). Plasmids were confirmed by PCR with the cloning primers above and fragments were gel purified and confirmed by Sanger sequencing with the forward primer.

## Lentivirus production, titration and cell line transduction

ACE2 lentivirus was produced by co-transfection of HEK293T or Lenti-X HEK293T cells with the pCDH-ACE2 plasmid, VSV-G and Gag-Pol using Lipofectamine 2000 Reagent (Thermo-Fisher Scientific) or Xfect Transfection Reagent (Takara Bio) as per manufacturers' instructions. Supernatant was collected 2 days after transfection and centrifuged at 500 x g for 10 min, and this was concentrated using Amicon Ultra-15 Centrifugal Filter Units with 100 kDa cutoff (Merck Millipore) as per manufacturers' instructions. Lentivirus was titrated by serial dilution of lentivirus and incubating with 5000 HEK293T cells in 96 well plates with 8 μg/ml polybrene for 3 days followed by GFP detection by flow cytometry (BD Biosciences LSRFortessa). Transduction units (TU) per ml was calculated by percentage of 5000 cells that are GFP positive multiplied by the dilution. For example if 2 μl lentivirus gives 5% GFP positive cells, the TU/ml is $((5/100)^*5000)^*(1000/2) = 125,000$ TU/ml. The p24 equivalent (ng/ml) was measured using Lenti-X GoStix Plus (Takara Bio) as per manufacturers' instructions.

Cell lines were transduced by incubating with lentivirus and 8 μg/ml polybrene for 2–3 days followed by 5 μg/ml puromycin treatment until most resistant cells expressed GFP. Cells were then infected with SARS-CoV-2 at MOI 0.1 for 1 hr at 37˚C, cells were washed with PBS and media replaced. Culture supernatant was harvested at the indicated time points and titered by $CCID_{50}$ assay as described.

## Spike RBD-Fc recombinant protein expression and validation

RBD region of SARS-CoV-2 Spike protein (Genbank accession number: MN908947, residues 319–524) was cloned in-frame with an upstream IgGH leader sequence and downstream human IgG1 Fc open reading frame into the mammalian expression vector pNBF. The recombinant expression plasmid was transfected into ExpiCHO cells (ThermoFisher) cultured in ExpiCHO-S Expression Medium (Gibco) as per manufacturers' protocols. At 7 days post transfection, supernatant was harvested and Fc fusion protein purified using Protein-A affinity chromatography (Cytiva). Purified recombinant protein was concentrated by ultrafiltration using Amicon Ultra-10 filters (Millipore, USA). Purity of recovered proteins was analyzed using 10% Tris-glycine SDS-PAGE visualized by Coomassie Brilliant Blue staining (S1A Fig), confirming pure RBD-Fc protein. Indirect ELISA with established antibodies was used to validate RBD-Fc folding. Briefly, RBD-Fc protein (2 μg/ml) was coated on the ELISA plate at 4˚C overnight. Then the plates were blocked using PBS containing 0.05% Tween-20 and 5% KPL Milk Diluent/Blocking Solution Concentrate (SeraCare) at 37˚C for 1 h. Serial dilutions of recombinant mouse CR3022 (anti-SARS RBD) [42, 43]) or G4 (anti-MERS control antibody) [118] were added to the plate and incubated at 37˚C for 1 h. After 4 washes, plates were incubated with a 1:2500 dilution of a horse radish peroxidase (HRP)-conjugated goat anti-mouse secondary antibody (Thermofisher) for 1 h at 37˚C. After a final wash, plates were developed for 5 min using TMB Single Solution chromogen/substrate (Thermofisher) before the reaction was stopped by addition of 2N $H_2SO_4$. Absorbance at 450 nm was then read on a Spectramax 190 Microplate reader (Molecular Devices). This confirmed anti-SARS RBD recognized the recombinant RBD-Fc, suggesting authentic folding (S1B Fig).

## RBD:ACE2 flow cytometry-based binding assay

HEK293T cells expressing ACE2 were collected with TrypLE (ThermoFisher) and centrifuged at 500 x g for 5 min. Cells were resuspended in 1 μg/ml recombinant RBD-Fc in FACS buffer (PBS + 10% FBS) and incubated at 4˚C for 45 min on a rotor. Cells were washed 3 times in FACS buffer and resuspended in 2 μg/ml Alexa Fluor 594 AffiniPure Donkey Anti-Human IgG, Fcγ fragment specific (Jackson Immunoresearch) at 4˚C for 30 min on a rotor. Cells were

washed 3 times in FACS buffer and subjected to flow cytometry using LSRFortessa Flow Cytometer (BD Biosciences).

## SARS-CoV-2 spike RBD:ACE2 mutagenesis modeling

PyMOL v4.60 (Schrodinger) was used for mutagenesis of the crystal structure of SARS-CoV-2 spike receptor-binding domain bound with ACE2 from the protein data bank (6M0J) [115]. The side chain orientation (rotamer) with highest frequency of occurrence in proteins was chosen for all amino acid substitutions. Polar interactions (blue lines) or any interactions within 3.5Å (yellow lines) were shown for selected residues. The K417N and N501Y mutations in the UK variant (B.1.1.7) and RSA variant (B.1.351) were introduced [44].

## UV-inactivated SARS-CoV-2 vaccine preparation and mice immunization

SARS-CoV-2 in 120 ml of medium was inactivated with at least 7650 J/m$^2$ UVC in a UVC 500 Ultraviolet Crosslinker (Hoefer) using 6 well plates (3 ml per well). Virus was determined to be inactivated by incubation with Vero E6 cells for 4 days without development of CPE. UV-inactive SARS-CoV-2 was then partially-purified using a 20% sucrose cushion centrifuged at 175,000 x g for 3–4 hr using SW32Ti rotor (Beckman Coulter). The pellet was resuspended in PBS and virus washed and concentrated using Amicon Ultra-15 Centrifugal Filter Units with 100 kDa cutoff (Merck Millipore). For infectious SARS-CoV-2 immunization, virus was also concentrated using Amicon Ultra-15 Centrifugal Filter Units with 100 kDa cutoff (Merck Millipore). Mice were injected s.c. in the base of the tail with 100 μl of UV-inactive or infectious SARS-CoV-2 with 25 μg adjuvant in 50 μl made as described previously [119, 120]. A boost was given 5–6 weeks after prime, and mice were bled to measure serum neutralizing titers 4–9 weeks after boost.

## Neutralization assay

Mouse serum was heat inactivated at 56˚C for 30 min, serially diluted 3 fold and dilutions (50 μl) incubated with 100 CCID$_{50}$ SARS-CoV-2 (50 μl) for 2 hr at 37˚C in round bottom 96 well plates in RPMI1640 supplemented with 2% FCS, and penicillin, streptomycin and L-glutamine as described above. The mouse serum-virus mixtures (100 μl) were then added to wells of 96 well plates containing 10$^5$ Vero E6 cells in 100 μl of medium supplemented with 10% FCS. After 4 days cells were fixed and stained by adding 50 μl formaldehyde (15% w/v) and crystal violet (0.1% w/v) (Sigma-Aldrich) overnight. The plates were washed in tap water, dried overnight and 100 μl/well of 100% methanol added to dissolve the crystal violet and the optical density (OD) was read at 595 nm using a 96-well plate reader (Biotek Synergy H4). The 50% neutralizing titers were interpolated from OD versus dilution graphs. The OD for wells with cells plus serum but no virus was deemed to 100% neutralization (0% CPE), and the OD seen for wells with cells plus virus and naïve control serum was deemed to be 0% neutralization (>80% CPE).

## Intrapulmonary lentivirus transduction and SARS-CoV-2 infection

C57BL/6J, IFNAR$^{-/-}$ [121] (originally provided by P. Hertzog, Monash University, Melbourne, VIC, Australia) and IL-28RA$^{-/-}$ mice [84, 122] were kindly provided by Bristol-Myers Squibb [123] and bred in-house at QIMRB. The conditions the mice were kept are as follows: light = 12:12 h dark/light cycle, 7:45 a.m. sunrise and 7:45 p.m. sunset, 15 min light dark and dark light ramping time. Enclosures: M.I.C.E cage (Animal Care Systems, Colorado, USA). Ventilation: 100% fresh air, eight complete air exchange/h/rooms. In-house enrichment: paper

cups (Impact-Australia); tissue paper, cardboard rolls. Bedding: PuraChips (Able scientific) (aspen fine). Food: Double bagged norco rat and mouse pellet (AIRR, Darra, QLD). Water: deionized water acidified with HCl (pH = 3.2). All mice were female mice and were 8 weeks to 1 year old (age matched between groups, see S10 Fig for all mice ages) at the start of the experiment. Mice were anesthetized using isoflurane and 4 μl of 1% L-α-Lysophosphatidylcholine from egg yolk (Sigma Aldrich) in water was administered into the lungs via the intranasal route. After 1 hour mice were given an intrapulmonary inoculation of approximately $2 \times 10^4$ TU of hACE2-pCDH lentivirus in 50 μl, and 1 week later received an intrapulmonary challenge with $10^5$ CCID$_{50}$ SARS-CoV-2 delivered via the intranasal route in 50 μl. Mice were sacrificed by cervical dislocation at day 2, 4 or 6 and lungs were collected. Right lung was immediately homogenized in tubes each containing 4 beads twice at 6000 x g for 15 seconds, and used in tissue titration as described above. Left lung or brain was placed in RNAprotect Tissue Reagent (QIAGEN) at 4˚C overnight then stored at -80˚C.

K18-hACE2 mouse (strain B6.Cg-Tg(K18-ACE2)2Prlmn/J, JAX Stock No: 034860) [6] were purchased from The Jackson Laboratory, USA, and bred and maintained in-house at QIMRB as heterozygotes by crossing with C57BL/6J mice. Mice were genotyped using Extract-N-Amp Tissue PCR Kit (Sigma Aldrich) according to manufacturers' instructions with the following primers; Forward 5'-CTTGGTGATATGTGGGGTAGA-3' and Reverse 5'-CGCTTCATCTCCCACCACTT-3' (recommended by NIOBIOHN, Osaka, Japan). Thermocycling conditions were as follows; 94˚C 3 min, 35 cycles of 94˚C 30 s, 55.8˚C 30 s, 72˚C 1 min, and final extension of 72˚C 10 min. hACE2 positive mice were infected with $5 \times 10^4$ SARS-CoV-2 i.n. as above and lungs RNA was harvested at day 4 for RNA-seq.

## Histology and immunohistochemistry

Lungs were fixed in 10% formalin, embedded in paraffin, and sections stained with H&E (Sigma Aldrich). Lung histopathology was scored blinded by examining whole lung sections and evaluating infiltration, bronchiole membrane sloughing, occluded bronchioles, bronchiolar oedema, and smooth muscle hyperplasia as criteria for disease [57]. Areas with infiltrate lesions greater than approximately 1500 μm were counted and measured using Aperio ImageScope software (Leica Biosystems, Mt Waverley, Australia) (v10). Bronchiolar membrane sloughing was determined when there was evidence of bronchiole epithelial cells shedding from the membrane into the lumen as previously shown [124, 125]. Occluded bronchioles were determined based on substantially reduced space within the bronchiole, either due to oedema or complete collapse, indicating occlusion. Bronchioles with oedema were counted when evidence of fluid was present within the bronchiole. Smooth muscle hyperplasia was counted based on a muscle expansion greater than the negative control. Automatic quantitation of white space was undertaken using QuPath v0.2.3 [126].

For hACE2 and SARS-CoV-2 immunohistochemistry, sections were affixed to positively charged adhesive slides and air-dried overnight at 37˚C. Sections were dewaxed and rehydrated through xylol and descending graded alcohols to Tris buffered saline (TBS) pH 7.6. Endogenous peroxidase activity was blocked by incubating the sections in 0.5% hydrogen peroxide in methanol for 10 minutes and slides rinsed three times in distilled water. Sections were transferred to Dako Epitope Retrieval Buffer (pH 9.0 for hACE2, pH 6.0 for SARS-CoV-2) and subjected to heat antigen retrieval (98˚C for 20 min for hACE2, or 125˚C for 5 min for SARS-CoV-2) using the Biocare Medical de-cloaking chamber, and slides allowed to cool for 20 minutes before transferring to TBS. For hACE2, Biocare Medical Background Sniper + 2% BSA was applied for 15 minutes to stop non-specific binding. For SARS-CoV-2 donkey anti-mouse Fab fragments (Jackson Immunoresearch) diluted 1:50 in Biocare Medical Rodent block M for

60 minutes. Recombinant rabbit anti-hACE2 antibody (Abcam, ab108209, EPR4436) diluted 1:200 in Biocare Medical Van Gough Yellow diluent, or mouse anti-SARS-CoV-2 spike monoclonal antibody 1E8 (Hobson-Peters *et al.* in preparation) diluted 1:70 in Biocare Medical Da Vinci Green diluent, was applied overnight at room temperature. Sections were washed three times, and Biocare Medical MACH2 Rabbit HRP Polymer was applied for 30 min at room temperature for hACE2, or goat anti-mouse HRP (Perkin Elmer) diluted 1:500 in 0.05% Tween 20 was applied at room temperature for 60 min for SARS-CoV-2. Sections were washed three times and color was developed in NovaRed (HRP) (Vector Laboratories) for 5 mins. Sections were washed in gently running tap water for 5–10 minutes to remove excess chromogen. Sections were lightly counterstained in Mayers' haematoxylin, and then dehydrated through ascending graded alcohols, cleared in xylene, and mounted using DePeX. All slides were scanned using Aperio AT Turbo (Aperio, Vista, CA USA) and analyzed using Aperio ImageScope software (Leica Biosystems, Mt Waverley, Australia) (v10).

## RT-qPCR

Mouse lungs were transferred from RNAlater to TRIzol (Life Technologies) and was homogenized twice at 6000 x g for 15 sec. Homogenates were centrifuged at 14,000 × g for 10 min and RNA was isolated as per manufacturers' instructions. cDNA was synthesized using ProtoScript II First Strand cDNA Synthesis Kit (New England Biolabs) and qPCR performed using iTaq Universal SYBR Green Supermix (Bio-Rad) as per manufacturers' instructions with the following primers; SARS-CoV-2 E (WHO, Carite, Berlin) [127] Forward 5'- ACAGGTACGTTAA TAGTTAATAGCGT -3' and Reverse 5'- ATATTGCAGCAGTACGCACACA, hACE2 Forward 5'- GATCACGATTCCCAGGACG -3' and Reverse 5'- TCCGGCTGAACGACAA CTC -3', mRPL13a [128] Forward 5'- GAGGTCGGGGTGGAAGTACCA -3' and Reverse 5'- TGCATCTTGGCCTTTTCCTT -3'. PCR fragments of SARS-CoV-2 E, ACE2 and mRPL13a using the same primers as above were gel purified and 10-fold serial dilutions of estimated copy numbers were used as standards in qPCR to calculate copies in samples reactions. SARS-CoV-2 E and ACE2 copies were normalized by mRPL13a copy number in each reaction. qPCR reactions were performed in duplicate and averaged to determine the copy number in each sample.

RT-qPCR using RNA from mouse lung or brain samples was performed as above using primers (Integrated DNA Technologies) for mouse Ccl8 (Forward 5'- GGGTGCTGAAAAG CTACGAGAG -3' and Reverse 5'- GGATCTCCATGTACTCACTGACC -3'), Ccr3 (Forward 5'- CCACTGTACTCCCTGGTGTTCA -3' and Reverse 5'- GGACAGTGAAGAGAAAGAG CAGG -3') and Cd163 (Forward 5'- GGCTAGACGAAGTCATCTGCAC -3' and Reverse 5'- CTTCGTTGGTCAGCCTCAGAGA -3'). Log$_2$ fold change was calculated using the $2^{-\Delta\Delta Ct}$ method [129] using mRPL13a as the housekeeping gene as above.

## RNA-seq

TRIzol extracted lung RNA was treated with DNase (RNase-Free DNAse Set (Qiagen)) followed by purification using RNeasy MinElute Cleanup Kit (QIAGEN) as per manufacturers' instructions. RNA concentration and quality was measured using TapeStation D1K TapeScreen assay (Agilent). cDNA libraries were prepared using the Illumina TruSeq Stranded mRNA library prep kit and the sequencing performed on the Illumina Nextseq 550 platform generating 75bp paired end reads. Per base sequence quality for >90% bases was above Q30 for all samples. The quality of raw sequencing reads was assessed using FastQC [130] (v0.11.8) and trimmed using Cutadapt [131] (v2.3) to remove adapter sequences and low-quality bases. Trimmed reads were aligned using STAR [132] (v2.7.1a) to a combined reference that

included the mouse GRCm38 primary assembly and the GENCODE M23 gene model [133], SARS-CoV-2 isolate Wuhan-Hu-1 (NC_045512.2; 29903 bp) and the human ACE2 mouse codon optimized sequence (2418 bp). Mouse gene expression was estimated using RSEM [134] (v1.3.0). Reads aligned to SARS-CoV-2 and hACE2 were counted using SAMtools [135] (v1.9). Differential gene expression in the mouse was analyzed using EdgeR (3.22.3) and modelled using the likelihood ratio test, glmLRT().

## Analyses of K18-hACE2 and Ad5-hACE2 RNA-seq data

RNA-seq datasets generated from the Winkler et al. study [9], and Sun et al. study [53] were obtained from the Gene Expression Omnibus (GSE154104 and GSE150847 respectively) and trimmed using Cutadapt (v2.3). Trimmed reads were aligned using STAR (v2.7.1a) to a combined reference that included the mouse GRCm38 primary assembly and the GENCODE M23 gene model, SARS-CoV-2 isolate Wuhan-Hu-1 (NC_045512.2; 29903 bp) and the human ACE2 transcript variant 1 (NM_001371415.1; 3339 bp). Mouse gene expression was estimated using RSEM (v1.3.0). Reads aligned to SARS-CoV-2 and hACE2 were counted using SAMtools (v1.9). Differential gene expression in the mouse was analyzed using EdgeR (3.22.3) and modelled using the likelihood ratio test, glmLRT().

## Pathway analysis

Up-Stream Regulators (USR), Diseases and Functions and canonical pathways enriched in differentially expressed genes in direct and indirect interactions were investigated using Ingenuity Pathway Analysis (IPA) (QIAGEN).

## Network analysis

Protein interaction networks of differentially expressed gene lists were visualized in Cytoscape (v3.7.2) [136]. Enrichment for biological processes, molecular functions, KEGG pathways and other gene ontology categories in DEG lists was elucidated using the STRING database [137] and GO enrichment analysis [138].

## Gene set enrichment analysis

Preranked GSEA [139] was performed on a desktop application (GSEA v4.0.3) (http://www.broadinstitute.org/gsea/) using the "GSEAPreranked" module. Gene sets from the supplemental materials from the Winkler et al. [9], Blanco-Melo et al. [64] and Wu et al. [65] (filename. GMT) studies were investigated for enrichment in our pre-ranked all gene list for SARS-CoV-2 infected day 2 mouse lung comparing plus versus minus ACE2 (S2B Dataset). Li et al. [89] blood transcription modules (BTM_for_GSEA_20131008.gmt, n = 346) was also used in GSEA to determine enrichment in our 'immunized' versus unvaccinated mouse lung at day 2 (S5B Dataset). The complete Molecular Signatures Database (MSigDB) v7.2 gene set collection (31,120 gene sets) (msigdb.v7.2.symbols.gmt: https://www.gsea-msigdb.org/gsea/msigdb/download_file.jsp?filePath=/msigdb/release/7.2/msigdb.v7.2.symbols.gmt) was used to run GSEAs on pre-ranked gene list for plus versus minus ACE2 at day 2 (S1B Dataset), and day 6 versus day 2 (S2B Dataset). Gene ontology [140, 141] and Enrichr [142, 143]) web-tools were also consulted.

## Interferome

The indicated DEG lists were entered into Interferome (www.interferome.org) [144] with the following parameters: *In Vivo*, Mus musculus, fold change 2 (up and down).

## Statistics

Statistical analyses of experimental data were performed using IBM SPSS Statistics for Windows, Version 19.0 (IBM Corp., Armonk, NY, USA). The t-test was used when the difference in variances was <4, skewness was > -2 and kurtosis was <2. Otherwise, the non-parametric Kolmogorov–Smirnov test or Kruskal-Wallis test was used. Repeated Measures Two-Way ANOVA was used for cell line virus growth kinetics, with a term for days post infection. Statistics for areas of leukocyte infiltration (lesion count and area) modelled lesion counts with a Poisson generalised linear model, using the square root of the sum of lesions sizes for each mouse to weight each mean. Spearman correlation was used for correlating hACE2 and SARS-CoV-2 expression levels.

## Supporting information

**S1 Fig. RBD-Fc validation. A)** Recombinant SARS-CoV-2 spike RBD-Fc purity was analyzed on reducing SDS-PAGE followed by Coomassie staining and **B)** folding validated using recombinant mouse anti-SARS spike antibody (msCR3022) or mouse anti-MERS spike antibody (msG4) in ELISA.
(TIF)

**S2 Fig. RBD-Fc:ACE2 binding assay.** Flow cytometry gating of RBD-Fc binding (AlexaFlour 594, y-axis) to ACE2-transduced (GFP+, x-axis) cell lines. Values are given in the table for the total percentage of cells expressing GFP, and GFP+ cells with RBD-Fc bound. n = 2–3 per cell line, with an independent repeat showing similar results.
(TIF)

**S3 Fig. Replication of SARS-CoV-2 in hACE2-lentivirus transduced mouse cell lines.** Growth kinetics of SARS-CoV-2 over a three day time course in mock transduced or hACE2-lentivirus transduced 3T3 or AE17 cells infected at MOI = 0.1. Data is the mean of triplicate wells and error bars represent SEM. Images of cells were taken using an inverted light microscope at day 3 post-infection and are representative of triplicate wells.
(TIF)

**S4 Fig. Intrapulmonary administration of hACE2-lentivirus does not transduce mouse brain. A)** RT-qPCR of mouse lung and brain RNA using primers for hACE2 gene normalized to mRPL13a levels. Data is individual mice (n = 3 per group) and is expressed as RNA copy number ratio calculated using a standard curve. Horizontal line indicates cut-off for reliable detection. **B)** RT-qPCR of mouse lung and brain RNA on day 4 post-infection using primers for SARS-CoV-2 E gene normalized to mRPL13a levels. Data is for individual mice and is expressed as RNA copy number ratio calculated using a standard curve. Horizontal line indicates cut-off for reliable detection.
(TIF)

**S5 Fig. SARS-CoV-2 and hACE2 read counts in RNA-Seq data.** RNA-Seq was performed on mouse lung from the same mice as in 'Fig 2'. **A)** hACE2 read counts normalized to total read count. Data points below the horizontal dotted line had read counts of zero. **B)** SARS-CoV-2 read counts normalised to total read count. Data points below the horizontal dotted line had read counts of zero. **C)** SARS-CoV-2 read count normalised to hACE2 read count. Circle with asterisk had hACE2 read count of 0, so the value was set to the SARS-CoV-2 read count (not normalised). **D)** SARS-CoV-2 reads aligned to reference genome viewed in Integrative Genome Viewer (IGV). Mice with hACE2 displayed reads mapped across the entire genome, with higher counts for structural gene sub-genomic RNA as also evident in hACE2-adenoviral

vector transduced cell lines [64]. **E)** SARS-CoV-2 read counts normalised to total mouse reads from RNA-seq data for lentivirus-hACE2 transduced mice at day 2, K18-hACE2 transgenic mice at day 4, and downloaded data for Ad5-hACE2 transduced mice at day 2 [53]. Statistics by Kolmogorov Smirnov test. **F)** hACE2-lentivirus reads aligned to the lentivirus reference genome viewed in IGV. RNA from hACE2-lentivirus transduced mouse lung had reads mapping across the entire hACE2 gene, but did not have reads mapping across GFP. **G)** hACE2 read counts normalised to total mouse reads from RNA-seq data for lentivirus-hACE2 transduced mice at day 2, K18-hACE2 transgenic mice at day 4, and downloaded data for Ad5-hACE2 transduced mice at day 2 [53]. Statistics by Kolmogorov Smirnov test.
(TIF)

**S6 Fig. SARS-CoV-2 titer and RNA correlate with hACE2 RNA levels. A)** Correlation of SARS-CoV-2/mRPL13a copy number ratio from Fig 2E with hACE2/mRPL13a copy number ratio from Fig 2C). Correlation is significant by Spearman correlation. **B)** Correlation of SARS-CoV-2 lung tissue titer from Fig 2D with hACE2/mRPL13a copy number ratio from Fig 2C). Correlation is significant by Spearman correlation. **C)** RT-qPCR of mouse lung RNA using primers for SARS-CoV-2 normalized to hACE2 introduced by lentivirus transduction. Data is individual mice from Fig 2E normalised to Fig 2C, and is expressed as RNA copy number calculated against a standard curve for each gene.
(TIF)

**S7 Fig. Additional anti-hACE2 IHC images.** Examples of anti-hACE2 IHC (brown staining) in mouse lung alveolar epithelial cells, additional to Fig 3G. Staining was evident in mouse lungs transduced with hACE2-lentivirus, and no staining was evident in mock transduced mouse lung.
(TIF)

**S8 Fig. Histological features that were not significantly affected by SARS-CoV-2 infection. A)** Occluded bronchioles were determined based on substantially reduced space within the bronchiole, either due to oedema or complete collapse, indicating occlusion. The percentage of bronchioles occluded as a proportion of total bronchioles for each mice is shown. When all hACE2-transduced mice are compared to mock transduced mice, both of which received an intra-lung inoculum of SARS-CoV-2, the percentage of bronchioles occluded approaches significance by Kruskal-Wallis test. **B)** Bronchioles with oedema were counted when evidence of fluid was present within the bronchiole. The percentage of bronchioles with oedema as a proportion of total bronchioles for each mice is shown. **C)** Smooth muscle hyperplasia was counted based on a muscle expansion greater than naïve mouse lung. Only 1 bronchiole (day 6 + hACE2-lentivirus) of all lungs examined had clear evidence of smooth muscle hyperplasia. **D)** Automatic quantitation of white space was undertaken using QuPath v0.2. n = 3–4 per group for all analyses.
(TIF)

**S9 Fig. RT-qPCR using Ccl8, Ccr3 and Cd163 primers and histological analyses indicate no significant inflammatory responses from intrapulmonary hACE2-lentivirus transduction at the time of harvest for RNA-Seq analyses. A)** $Log_2$ fold change for Ccl8, Ccr3 and Cd163 determined by RNA-Seq (S1C Dataset) is represented by the black bar. $Log_2$ fold change for Ccl8, Ccr3 and Cd163 was calculated from RT-qPCR data using the $2^{-\Delta\Delta Ct}$ method for the indicated comparisons. RT-qPCR for Ccl8, Ccr3 and Cd163 using the same RNA samples used in RNA-Seq validates both assays. Statistics are by t-test and shows significantly higher induction of Ccl8 in SARS-CoV-2 infected mice (green) compared to LPC plus vehicle only (purple), and significantly higher induction of Ccr3 and Cd163 compared to

hACE2-lentivirus transduction only (blue and red). **B)** H&E stained lung sections from naïve mice (left) and mice that received intrapulmonary LPC + hACE2-lentivirus harvested at 9 days post-inoculation (right).
(TIF)

**S10 Fig. Mouse ages.** Individual mouse ages are shown in days at the time of SARS-CoV-2 inoculation.
(TIF)

**S1 Dataset. RNA-Seq gene lists and downstream bioinformatic analyses of responses at day 2 post-infection.**
(XLSX)

**S2 Dataset. RNA-Seq gene lists and downstream bioinformatic analyses of responses at day 6 post-infection.**
(XLSX)

**S3 Dataset. RNA-Seq gene lists and downstream bioinformatic analyses of responses in IFNAR$^{-/-}$ versus C57BL/6J mice at day 2 post-infection.**
(XLSX)

**S4 Dataset. RNA-Seq gene lists and downstream bioinformatic analyses of responses in IL-28RA$^{-/-}$ versus C57BL/6J mice at day 2 post-infection.**
(XLSX)

**S5 Dataset. RNA-Seq gene lists and downstream bioinformatic analyses of responses in immunized versus unvaccinated mice at day 2 post-infection.**
(XLSX)

## Acknowledgments

From QIMR Berghofer MRI we thank Dr I Anraku for managing the PC3 (BSL3) facility, Dr Clay Winterford for the histology and immunohistochemistry, animal house staff for mouse breeding and agistment, and Dr. Gunter Hartel for statistical advice. We thank Dr Alyssa Pyke and Mr Fredrick Moore (Queensland Health, Brisbane) for providing the SARS-CoV-2 isolate. We thank Dr. David Harrich for providing the lentivirus vector system. We thank Dr. Simon Phipps for providing the IL-28RA$^{-/-}$ mice. We thank Monash Genome Modification Platform for providing the plasmid containing the mouse-codon optimized human ACE2 gene.

## Author Contributions

**Conceptualization:** Daniel J. Rawle.

**Data curation:** Daniel J. Rawle, Troy Dumenil, Andreas Suhrbier.

**Formal analysis:** Daniel J. Rawle, Troy Dumenil, Wilson Nguyen, Andreas Suhrbier.

**Funding acquisition:** Daniel J. Rawle, Andreas Suhrbier.

**Investigation:** Daniel J. Rawle, Thuy T. Le, Kexin Yan, Bing Tang, Daniel Watterson, Naphak Modhiran, Jody Hobson-Peters.

**Methodology:** Daniel J. Rawle, Andreas Suhrbier.

**Project administration:** Daniel J. Rawle, Andreas Suhrbier.

**Resources:** Daniel Watterson, Naphak Modhiran, Jody Hobson-Peters, Andreas Suhrbier.

**Supervision:** Daniel J. Rawle, Andreas Suhrbier.

**Validation:** Daniel J. Rawle.

**Visualization:** Daniel J. Rawle, Troy Dumenil, Daniel Watterson, Naphak Modhiran, Cameron Bishop, Andreas Suhrbier.

**Writing – original draft:** Daniel J. Rawle.

**Writing – review & editing:** Daniel J. Rawle, Andreas Suhrbier.

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
