## [Decision Letter · Decision Letter 0]

14 Apr 2021

Dear Rawle,

Thank you very much for submitting your manuscript "ACE2-lentiviral transduction enables mouse SARS-CoV-2 infection and mapping of receptor interactions" for consideration at PLOS Pathogens. As with all papers reviewed by the journal, your manuscript was reviewed by members of the editorial board and by several independent reviewers. In light of the reviews (below this email), we would like to invite the resubmission of a significantly-revised version that takes into account the reviewers' comments.

While there is varying degrees of enthusiasm for the lentiviral transduction model presented, all three reviewers commented on the need for additional studies that characterize the type of cells transduced by the lentiviral constructs in the authors' mouse model. We also ask that the authors address the other reviewers' comments constructively.  

We cannot make any decision about publication until we have seen the revised manuscript and your response to the reviewers' comments. Your revised manuscript is also likely to be sent to reviewers for further evaluation.

Sincerely,

Benhur Lee

Section Editor

PLOS Pathogens

Benhur Lee

Section Editor

PLOS Pathogens

Kasturi Haldar

Editor-in-Chief

PLOS Pathogens

orcid.org/0000-0001-5065-158X

Michael Malim

Editor-in-Chief

PLOS Pathogens

orcid.org/0000-0002-7699-2064

Reviewer's Responses to Questions

**Part I - Summary**

Reviewer #1: This study describes a lentiviral transduction system for the expression of hACE2 in cell lines and mice. The utility of the model is clearly displayed for evaluating ACE2 residues important for in vitro replication, and the structure models are presented clearly. The study is well-written and the results are clearly described. However, there is a heavy emphasis on gene expression analysis that is of questionable relevance to human disease given the limited replication observed in the model.

Reviewer #2: This manuscript by Rawle et al., describes evaluation of a lentiviral vector encoding human ACE2 receptor as a means to sensitize diverse mouse strains to infection with human SARS-CoV-2 viruses. The approach is a useful contribution to the field which will complement existing studies which have evaluated transgenic hACE2 mouse models, and other viral vector mediated models in which hACE2 can be transiently delivered to any mouse strain or genetic knockout using adenoviral or adeno-associated viruses. Each model has its pros and cons. Therefore, detailed descriptions of the utility of each model will be informative to investigators in the field. This manuscript has shown that lentiviral delivery of hACE2 to mouse lung is an effective means of sensitizing C57Bl6, IFNAR-/- and IL-28RA-/- mice to SARS-CoV-2 infection. The authors also shed light on the residues within the SARS-CoV-2 receptor binding domain (RBD), as well as human and murine ACE2 which contribute to the specificity of binding and ability to use ACE2 as an entry receptor. In addition, the authors provide a comprehensive evaluation of genes which are differentially expressed following challenge in mice, and compare these data sets with existing literature for both the K18-hACE2 transgenic mouse model, and human data sets. This is a nice validation of the model.

In general, the manuscript is easy to follow, however there are several sections which require clarification and additional detail. Some jargon and acronyms for gene expression data could be clarified to make the figures more accessible, and to make the context apparent to non-expert readers. There are some discrepancies in figures which need to be addressed and additional experiments are needed. The model would benefit from the addition of histology to confirm the spatial distribution of hACE2 expression following lentivirus transduction, as well as H&E to visualize lung pathology post-challenge. These are important parameters for establishing the model when compared with existing studies.

Reviewer #3: Andreas Suhrbier and colleagues submitted a well written manuscript that details the development of a lentivirus expression system for ACE2. Lentivirus transduction of mouse airway with human ACE2 conferred susceptibility to infection with SARS-CoV-2. However, infection of the lentivirus transduced mice resulted in minimal body weight loss. This observation suggests that although gene delivery of hACE2 promoted virus replication resulting in detectable infectious virus titers, the level of virus replication was still restricted. Immunohistochemistry (IHC) to detect transduced GFP or hACE2 and viral antigen would provide supportive data for the limited body weight changes. These IHC data would also help with interpretation of the RNAseq datasets in terms of location of viral replication and infected cells. The authors identify four amino acid residues within mouse ACE2 that are necessary for infection by SARS-CoV-2; however, the authors did not experimentally examine if the changes altered binding affinity. The investigative team utilized their model system to examine the role of type I and type III interferon pathways in respiratory disease resulting from SARS-CoV-2 infection. Although the lack of either interferon pathway did not alter the kinetics or magnitude of virus replication, the authors did note from their RNA-seq data differences in differentially expressed genes, which suggests that these interferon pathways target distinct subsets of genes.

**Part II – Major Issues: Key Experiments Required for Acceptance**

Reviewer #1: Major points

1) What cells does the lentivirus transduce in vivo? If this is known in the literature it should be referenced. If not staining for hACE2 expression in the lung should be included. Also does the lentivirus induce any measurable inflammation and at what time points?

2) Experiments in Fig 1B need to be repeated at least one time and preferably 2 to determine if they are reproducible. Similarly, some of the groups in figure 2 only have 2 animals, this is problematic given the large spread in the results. RNA copy number does not correlate well at day 2 when replication is detected, this is surprising given how consistent hACE2 expression levels are, so some explanation should be provided for these results.

3) In my opinion the differential gene expression analysis doesn’t warrant 3 full figures. While there is some useful information here, because viral replication in this model is limited to the few cells that take up the lentivirus, the relevance of this data to human disease is not clear. Also given that replication is limited to D2 post-infection, its again unclear the relevance of the D6 tissue repair signatures. Likewise, loss of IFN-related gene upregulation in IFNAR-deficient mice is not surprising. These experiments would have been better controlled by transducing the control group with empty or irrelevant lentivirus, to control for any inflammation induced by the lentivirus or lysophosphatidylcholine.

Reviewer #2: Major comments

• Knowledge regarding the anatomical distribution of expression of hACE2 delivered by a lentiviral vector in the mouse lung would be informative to the field. As this study aims to contribute the utility of this SARS-CoV-2 sensitization model, information regarding the histological expression of hACE2 would be informative to investigators in deciding which model is most appropriate to their study. In addition to adding histology (IHC staining for hACE2), the inclusion of H&E stained lung sections after SARS-CoV-2 challenge would demonstrate the “severity” of the model, and could verify the potential effect of inflammatory cell infiltration diluting RT-qPCR results (as stated on Page 13, Line 327-329 with regard to Fig.6C). The authors state on Page 10, Line 239 that severe disease in SARS-CoV-2 animal models is typically measured by weight loss, however this is not always the case as many models do not result in weight loss. Again, this supports the need to provide histological specimens to determine lung pathology/severity for this model.

• It is unclear if some of the same mouse groups have been used for data presented in Fig.2C and Fig.6C, untransduced and unvaccinated. The data points look very similar. Also, the same observation has been made for Fig.2E and Fig.6D for the same groups. If these were the same groups run for different arms of an experiment, this should be clearly stated in the text and in the figure legends.

• More detail is required for the animal experiments. For example, group sizes are not defined and in Fig. 2, it appears that only an n=2 were analyzed for the C57Bl6 mice. Age and sex of animals should also be clearly outlined for routine strains and genetic knockouts.

Reviewer #3: The authors should perform immunohistochemistry to examine the pulmonary distribution of hACE2 gene delivery and assess the type and distribution of infected pulmonary cells.

**Part III – Minor Issues: Editorial and Data Presentation Modifications**

Reviewer #1: Minor points

1) Given that many of the mutations do not result in the predicted phenotypes in terms of increased/decreased replication, the results should be further discussed and placed in the context of the literature in terms of what is known about mouse adaptation and variants.

2) You claim that T92Q doesn’t result in any change in SARS-CoV-2 replication, however there is no validation that the mutation results in a glycosylation change in these cells.

3) The explanation of Fig 2F is confusing and I’m not sure the point you are trying to make. It may be more useful to present read numbers of hACE2 and SARS-CoV-2 in the different systems to see how they compare.

4) Ln 351 – you claim a key advantage of lentiviruses is long term stable expression - while adenovirus does not integrate, it does induce long-term expression

Reviewer #2: Minor comments

• The authors should justify why an interval of 1 week between lenti-hACE2 administration and challenge was selected. Is this known to be the maximal peak for transgene expression following lung delivery?

• What is known about lentivirus off target effects following intranasal administration. When comparing to the K18-hACE2 model this would be useful to know. Are lentiviruses known to infect olfactory bulb or to gain access to the brain?

• Page 6, Line 148-155. The authors should add some comment on the role of various proteases ie TMPRSS2, and their distribution in different cell lines, in affecting the ability for SARS-CoV-2 to replicate (PMID: 32142651).

• It is not always explicitly clear if untransduced animals were also challenged with SARS-CoV-2. This should be more clearly stated throughout the manuscript (although it is noted that this is outlined on Page 8, Line 193).

• Figure.2F. No description in the methods is provided for the use of the Ad5-hACE2 transduction model as a comparator.

• Figure.3. A more detailed explanation of what is being analysed in 3D should be described in the figure legend. In addition, please indicate in Page 8, Line 205-206 if this was performed on SARS-CoV-2 infected lung.

• A section describing the limitations of this model in comparison to others should be discussed. Page 3, Lines 75-78. Please provide specific information regarding the duration of expression, and the specific cell types transduced based on the existing literature. Putting this into context by comparing and contrasting against existing models for SARS-CoV-2 (ie. Adeno or AAV) would be useful to the field. It would also be informative to comment on if anti-vector immunity affects transgene expression when delivered by a lentivirus, or if lentiviral delivery is associated with any inflammation.

• Page 3, Line 63-65 and Lines 72-74. These statements could be updated with some discussion regarding a recent pre-print which suggests that some SARS-CoV-2 variants can infect mice using murine ACE2. This article also contains some investigation of binding residues for ACE2, so some interpretation and discussion of these data would be informative. I appreciate this pre-print is a new addition to the literature and was uploaded after this manuscript https://www.biorxiv.org/content/10.1101/2021.03.18.436013v1.full.pdf.

• Page 4, Line 80. Define GM.

• The statement on Page 7, Line 166-167 should be revised as the SARS-CoV-2 challenge doses in citations 10 and 43 differed 10-fold which may have contributed to differences in weight loss between the reported studies.

• Page 11, Line 277-279. This statement should be elaborated on. Prior studies also showed that inhibition of Type I IFN did not affect SARS-CoV-2 replication but that it did affect the inflammatory response and resulted in pathological signs of severe viral pneumonia. The addition of histological data (as proposed in comments above) would add support to this for the lentivirus-hACE2 model.

• Page 14, Line 351-353. The authors should explain why long-term expression would be preferable and what sort of in vivo studies in mice would benefit from this.

Reviewer #3: There are no minor issues of concern.

PLOS authors have the option to publish the peer review history of their article (what does this mean?). If published, this will include your full peer review and any attached files.

Reviewer #1: No

Reviewer #2: No

Reviewer #3: No
---

## [Editor Report · Decision Letter 1]

17 Jun 2021

Dear Rawle,

We are pleased to inform you that your manuscript 'ACE2-lentiviral transduction enables mouse SARS-CoV-2 infection and mapping of receptor interactions' has been provisionally accepted for publication in PLOS Pathogens.

Best regards,

Benhur Lee

Section Editor

PLOS Pathogens

Benhur Lee

Section Editor

PLOS Pathogens

Kasturi Haldar

Editor-in-Chief

PLOS Pathogens

orcid.org/0000-0001-5065-158X

Michael Malim

Editor-in-Chief

PLOS Pathogens

orcid.org/0000-0002-7699-2064

Reviewer Comments (if any, and for reference): The authors have made a good faith effort to answer the major criticisms raised by all three reviewers. The addition of immhohistochemical data identifying the cell types that are  infected in vivo, combined with more biochemical analysis makes this model of utility to the field.

---

## [Editor Report · Acceptance letter]

29 Jun 2021

Dear Rawle,

We are delighted to inform you that your manuscript, "ACE2-lentiviral transduction enables mouse SARS-CoV-2 infection and mapping of receptor interactions," has been formally accepted for publication in PLOS Pathogens.

Best regards,

Kasturi Haldar

Editor-in-Chief

PLOS Pathogens

orcid.org/0000-0001-5065-158X

Michael Malim

Editor-in-Chief

PLOS Pathogens

orcid.org/0000-0002-7699-2064